# Structure is Supervision:
# Multiview Masked Autoencoders for Radiology

**Sonia Laguna**\*                                                              *slaguna@inf.ethz.ch*
**Andrea Agostini**\*                                                           *anandrea@inf.ethz.ch*
**Alain Ryser**
**Samuel Ruiperez-Campillo**
**Irene Cannistraci**
**Moritz Vandenhirtz**
*Department of Computer Science, ETH Zurich*

**Stephan Mandt**
*Department of Computer Science, UC Irvine*

**Nicolas Deperrois**
**Farhad Nooralahzadeh**
**Michael Krauthammer**
*Department of Quantitative Biomedicine, University of Zurich*

**Thomas M. Sutter**[†]
**Julia E. Vogt**[†] *Department of Computer Science, ETH Zurich*

**Reviewed on OpenReview:** *https://openreview.net/forum?id=wryiHLzieX*

## Abstract

Building robust medical machine learning systems requires pretraining strategies that exploit the intrinsic structure present in clinical data. We introduce Multiview Masked Autoencoder (MVMAE), a self-supervised framework that leverages the natural multi-view organization of radiology studies to learn view-invariant and disease-relevant representations. MVMAE combines masked image reconstruction with cross-view alignment, transforming clinical redundancy across projections into a powerful self-supervisory signal. We further extend this approach with MVMAE-V2T, which incorporates radiology reports as an auxiliary text-based learning signal to enhance semantic grounding while preserving fully vision-based inference. Evaluated on a downstream disease classification task on three large-scale public datasets, MIMIC-CXR, CheXpert, and PadChest, MVMAE consistently outperforms supervised and vision–language baselines. Furthermore, MVMAE-V2T provides additional gains, particularly in low-label regimes where structured textual supervision is most beneficial. Together, these results establish the importance of structural and textual supervision as complementary paths toward scalable, clinically grounded medical foundation models.

## 1 Introduction

Foundation models have accelerated progress in artificial intelligence for medicine (Moor et al., 2023; Pai et al., 2024) and pathology (Chen et al., 2024a; Lu et al., 2024). Yet, the data realities in healthcare differ sharply from the internet-scale corpora that power general-purpose vision–language systems. Within the radiology domain, chest radiography datasets such as MIMIC-CXR (Johnson et al., 2019a), CheXpert (Chambon et al., 2024), Chest X-ray Wang et al. (2017), and PadChest (Bustos et al., 2020) comprise hundreds of thousands—not billions—of studies, labels are expensive and often noisy (Gündel et al., 2021), and institutional acquisition protocols vary widely (Hassanzadeh et al., 2018; Chambon et al., 2024). As a result, purely supervised pipelines struggle to remain performant, and generic vision-language pretraining may underperform or require costly adaptation to domain benchmarks (Chaves et al., 2024). Self-supervised

---

\*These authors contributed equally. [†]Shared senior authorship.

learning (SSL) offers a path forward by reducing reliance on expert labels (Radford et al., 2021; Tiu et al., 2022b; Azizi et al., 2021). Compared to natural images, the data generation process for radiographs defines a clear structure: one or multiple scans, along with a corresponding radiology report, constitute a radiology study. Nevertheless, many medical SSL methods still treat radiographs of the same study as independent images (Ghesu et al., 2022; Tiu et al., 2022a), disregarding the structure of clinical exams.

In practice, radiology studies are structured: they frequently contain multiple image projections (e.g., frontal or lateral), repeated acquisitions, and an accompanying report. This structure encodes strong geometric (anatomical and physiological) and semantic coherence across views that radiologists routinely exploit for interpretation. Prior work has leveraged portions of such structures, showing improvements in robustness and label efficiency (Mo & Liang, 2024; Pellegrini et al., 2025; Chen et al., 2024b). However, reconstruction-only objectives are view-specific and ignore cross-view correspondence (Xiao et al., 2023); contrastive objectives ignore reconstruction quality (Nguyen et al., 2022); and vision–language approaches may depend on report availability and quality, emphasizing abstract, label-driven semantics rather than detailed spatial or visual correspondence within images (Pellegrini et al., 2025; Chen et al., 2024b). A scalable SSL pretraining recipe for chest radiography should harness both local visual detail and study-level consistency.

Following this intuition, we propose Multiview Masked Autoencoder (MVMAE), a study-centric pretraining framework that couples masked reconstruction with cross-view alignment to learn view-invariant representations without relying on external supervision, as illustrated in Figure 1. MVMAE treats the different image projections acquired within the same study as natural positive pairs and jointly optimizes: (i) a masked image modeling objective per view to capture the complementary information from multiple X-ray views acquired during a patient examination; and (ii) a study-level alignment objective to regularize embeddings across projections of the same medical study. This converts clinical redundancy into self-supervision. Reflecting clinical practice, we adopt a study-level organization and a unified policy that retains single and multi-view exams, incorporates uncertain views, and aggregates multiple projections within a study. This approach avoids over-counting, preserves the internal structure radiologists actually use, and allows us to probe how performance scales with the number of available projections at test time. Although we focus on chest-X-rays, the same strategy could extend to any life-science domain that offers repeated or complementary scans, such as longitudinal magnetic resonance imaging (MRI) or multi-sequence computed tomography (CT). Additionally, no open dataset fully captures alone the variability of chest radiography (Zech et al., 2018). Institutional protocols, patient demographics, and acquisition devices differ substantially between collections such as MIMIC-CXR, CheXpert, Chest X-Ray, and PadChest. It is therefore crucial to pretrain jointly on all four datasets, allowing the model to internalize the variability inherent to clinical imaging and produce representations that transfer effectively across institutions.

To further inject semantic grounding when reports are available, we introduce MVMAE-V2T, which augments MVMAE with a lightweight vision-to-text objective. Reports are used only during pretraining as an auxiliary signal in addition to image-based objectives; at test time, the model remains a vision-only encoder. This design contrasts with CLIP-style (Zhang et al., 2025) or instruction-tuned radiology Vision-Language Models (VLMs) (Chen et al., 2024b; Deperrois et al., 2025) that base their objectives on image-text alignment or text generation only.

**Contributions** Our work advances multimodal self-supervised learning for radiology along four main directions: (i) We introduce MVMAE, the first multi-view masked autoencoder for radiology that adapts established masked reconstruction and cross-view alignment paradigms to the structure of medical data, yielding view-invariant yet detail-rich visual representations and consistently outperforming supervised and vision–language baselines, while providing better calibrated predictions. (ii) We extend this framework into MVMAE-V2T, which incorporates radiology reports as auxiliary supervision during pretraining. The additional vision-to-text signal enhances semantic grounding and offers clear gains under limited labeled data, without requiring text at inference. (iii) We adopt a clinically realistic, study-centric evaluation policy that mirrors real-world diagnostic practice and allows analysis of how cross-view consistency scales with the number of projections. (iv) We integrate three large public datasets: MIMIC-CXR, CheXpert Plus, and PadChest, into a unified benchmark, demonstrating consistent improvements across institutions and analyzing the per-label behavior of our models. Together, these results establish MVMAE and MVMAE-V2T as a simple yet effective blueprint for scalable, structure-aware medical foundation models.

(a) Pretraining        (b) Finetuning

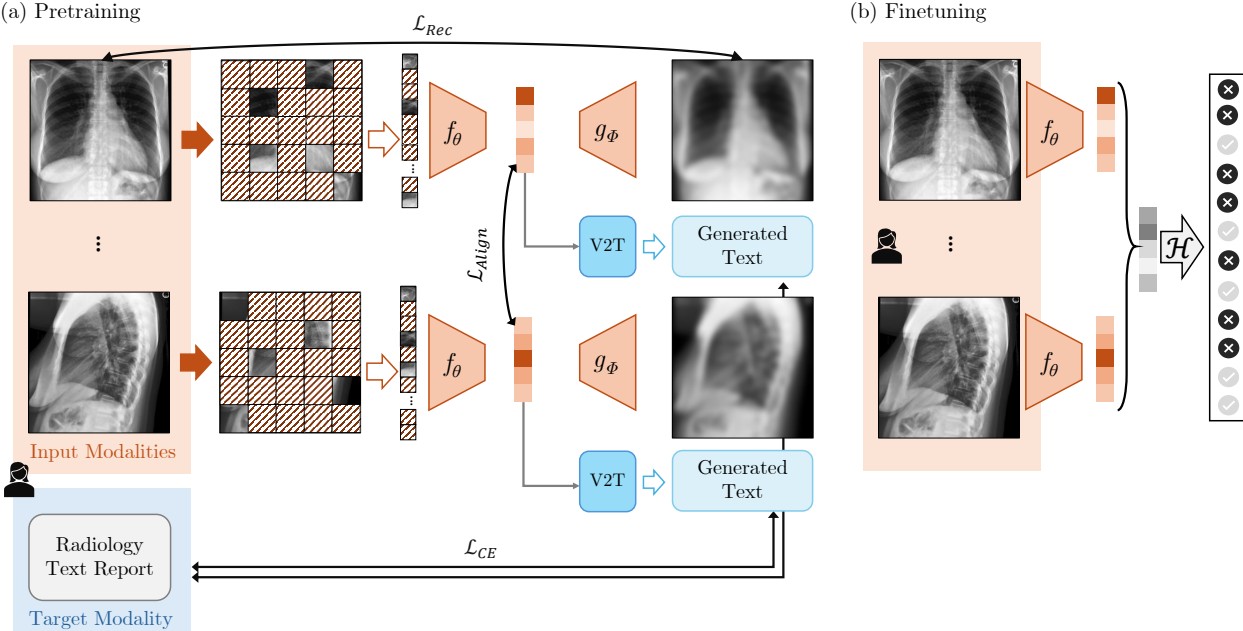

Figure 1: **Framework overview**. (a) Pretraining and (b) Finetuning stages of the proposed MVMAE and MVMAE-V2T frameworks. Each study includes multiple views processed jointly through masked reconstruction and cross-view alignment losses. MVMAE-V2T additionally incorporates a vision-to-text objective.

## 2 Related Work

**Self-supervised pretraining for medical imaging.** SSL has emerged as a key strategy for medical image representation learning, where expert annotation is costly and label distributions are imbalanced. Early works adapted general-purpose pretext tasks, such as context restoration or rotation prediction (Zhuang et al., 2019) to radiological images, followed by contrastive frameworks like SimCLR (Chen et al., 2020) and MoCo (He et al., 2020), which encourage invariance to data augmentations. Applied to chest radiographs, these methods demonstrated that large unlabeled datasets such as MIMIC-CXR (Johnson et al., 2019b) could yield strong feature encoders transferable to downstream classification and localization tasks (Azizi et al., 2021; Tiu et al., 2022b). However, these approaches typically treat each image as an independent instance, disregarding the inherent relational structure of radiology studies, such as co-acquired frontal and lateral projections. As a result, while they succeed in learning general visual semantics, they fail to exploit the intra-study coherence and anatomical consistency that characterize radiological data.

**Masked and reconstruction-based pretraining.** Masked image modeling has recently become the dominant paradigm in visual self-supervision. Masked Autoencoders (MAE) (He et al., 2022) learn dense representations by reconstructing masked patches of the input image, and domain-specific adaptations have been proposed for radiology (Xiao et al., 2023; Mo & Liang, 2024). These studies emphasize that optimal masking ratios and reconstruction targets differ from natural-image setups, and that subtle medical textures require careful tuning of pretext objectives. While MAEs capture fine-grained image statistics and low-level anatomy, their reconstruction targets remain view-specific and intra-image only. In the context of CXRs, they ignore a rich source of information: the geometric correspondence between frontal and lateral views of the same exam. In practice, each radiological study provides paired observations of the same anatomy under complementary projections. Yet, existing MAE pretraining works treat them as unrelated samples, forgoing a powerful form of implicit supervision.

**Multimodal and vision-language pretraining.** Paired image–report datasets have enabled cross-modal learning between radiographs and text. VLMs, such as MedCLIP, CheXagent, and RaDialog (Boecking et al., 2022; Pellegrini et al., 2025; Chen et al., 2024b; Deperrois et al., 2025), align visual and textual representations through contrastive or generative objectives, yielding transferable multimodal features. However, these approaches rely heavily on the availability and quality of reports, which vary across institutions, and the supervision they provide is semantic rather than geometric. In other words, text describes what is seen, but not how distinct radiographic views of the same patient relate anatomically to one another.

**Cross-view and longitudinal pretraining.** A smaller number of studies have begun to explore the structural redundancy in radiology data. Multi-view or correspondence-based contrastive methods (Nguyen et al., 2022; Zhou et al., 2023; Zeng et al., 2023; Agostini et al., 2025) exploit spatial and anatomical consistency across projections or timepoints, improving representation robustness and patient-level generalization. Still, these works remain limited either to contrastive pairing or simple view averaging; they do not yet integrate these constraints with reconstruction-based learning, nor do they exploit the joint information flow between modalities at scale.

In summary, prior medical pretraining approaches have successfully adapted self-supervised and multimodal objectives to radiological data but have largely abstracted away the structured nature of clinical imaging studies. Most treat radiographs as independent images, neglecting that each study inherently couples multiple views, repeated acquisitions, and textual context into a semantically and geometrically coherent whole.

Our work builds on this observation by explicitly leveraging the structure of chest radiography, paired frontal–lateral projections, as a self-supervisory signal. In contrast to previous view-agnostic reconstruction or report-dependent alignment, our method couples masked reconstruction with cross-view regularization and vision-to-text reconstruction, yielding representations that are both detail-preserving and view-invariant. This approach moves toward domain-adapted foundation models that learn not just from more data but from the structure of the data itself.

## 3 Dataset and Preprocessing

**MIMIC-CXR, CheXpert, PadChest, and Chest X-ray as multimodal resources.** We conduct the experiments on four large-scale publicly available chest radiograph archives: MIMIC-CXR (Johnson et al., 2019b), CheXpert Plus (Chambon et al., 2024), PadChest (Bustos et al., 2020), and Chest X-ray (Wang et al., 2017). MIMIC-CXR contains over 377,000 images from 227,000 studies linked to 65,000 patients. CheXpert Plus is an extension of the original CheXpert dataset, which includes more reliable labels, reports, and additional metadata, comprising over 224,000 images from 65,000 patients. PadChest includes approximately 110,000 studies comprising around 160,000 images from roughly 67,000 patients. For a detailed overview of the datasets, please refer to Table 5 in Appendix A. All include routine chest radiographs and reports acquired in critical-care settings.

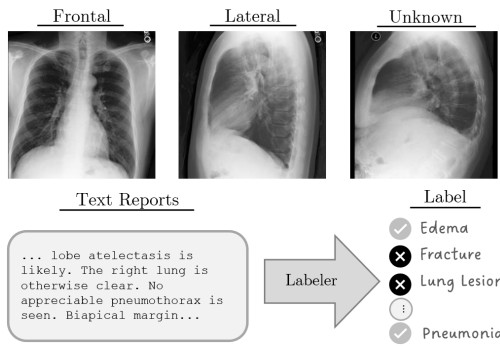

**Study structure.** These three corpora are organized at the *study* level, where each study groups together one or more radiographic projections associated with a single radiology report. Image quality varies substantially across datasets due to differences in patient positioning, acquisition devices, and clinical conditions, providing a realistic testbed for robust pretraining. Finally, the Chest X-ray dataset comprises roughly 109,000 frontal-view images from approximately 31,000 patients. Since the original text reports are not publicly released and the label ontology differs from those of the other datasets, we use it only for unsupervised pretraining, rather than for downstream classification or evaluation. This setup broadens visual diversity during pretraining while preserving label consistency across evaluation datasets.

Figure 2: **Study-level instance in MIMIC-CXR.** The first row shows three different views in the same study, with their corresponding report and final labels.

We treat a *study* , the complete set of projections acquired during a patient encounter, as the fundamental unit of analysis. Projections are grouped into two main families: *frontal* (posterior–anterior *PA* or anterior–posterior *AP*), and *lateral* (left-lateral *LL* or generic lateral), with a subset labeled as *unknown*, i.e., undefined in the metadata or view categories of low prevalence. To capture their multiview nature, we adopt a unified *FLU* (Frontal–Lateral–Unknown) policy that includes all studies containing at least one valid projection, indexing each available view as $v \in \{f, l, u\}$. Formally, each dataset yields a collection $\mathbf{X}^{(i)} = \left\{ \mathbf{x}_v^{(i)} \right\}_{v \in \mathcal{V}^{(i)}}$, with $\mathcal{V}^{(i)} = \{v_1, \ldots, v_{n^{(i)}}\}$ denoting the set of view types present in study $i$, $n^{(i)} \geq 1$ being the number of views per study $(i)$. Please note that we allow multiple instances of the same view

type $v$. This ensures inclusivity: single-view studies are retained, multi-view studies contribute all available projections, and uncertain view types are incorporated rather than discarded. When available, reports are also associated with each study and used to derive the final disease prediction labels. An illustrative example of what one study unit comprises, including its multiple projections, corresponding report, and final label set, is shown in Figure 2. Extended information about the dataset distribution, illustrations, and study nature can be found in Appendix A.

**Preprocessing pipeline and data split.** All radiographs are resized to $224 \times 224$. For MIMIC-CXR and CheXpert, study-level labels are inherited from the CheXpert labeler (Irvin et al., 2019), covering 14 diagnostic categories. Following Haque et al. (2023), the three non-positive states ("negative", "not mentioned", "uncertain") are collapsed into a single 0-class, treating only explicit positives as 1. In MIMIC-CXR and CheXpert, we use the official training, validation, and test splits. Since PadChest and Chest X-ray do not provide predefined splits, we create them manually, stratifying by patient to prevent leakage and to obtain a data-split ratio comparable to MIMIC-CXR and CheXpert. For PadChest, we allocate 97% of studies to training and split the remaining 3% evenly between validation and test. For Chest X-ray, we combine the original training and validation sets and use 98% for training and 2% for validation, while keeping the official test set unchanged. For PadChest, additional preprocessing was required. The dataset originally includes 193 labels, spanning radiographic findings, differential diagnoses, and anatomical locations. To ensure comparability, we mapped these labels into the same 14-category space defined by the CheXpert labeler. This mapping was constructed automatically with the assistance of Claude Code 2.0.35 (Anthropic, 2025) to identify the closest and most clinically consistent correspondences between PadChest labels and the CheXpert label set. This harmonization enables a unified evaluation across all three datasets. Additional details on the datasets and preprocessing, as well as statistics on the labels and other relevant information, are provided in Appendix A.

# 4 Methods

We study chest X-ray representation learning across multiple pretraining regimes, with a focus on the Multi-view Masked Autoencoders, MVMAE[1] , which combines masked reconstruction with a cross-view alignment loss to exploit study-level structure, and can be extended with text supervision, resulting in MVMAE-V2T. An overview of our approach is shown in Figure 1. For evaluation purposes, we investigate ablated variants of MVMAE and state-of-the-art vision–language models pretrained on broader medical data.

## 4.1 MVMAE: Multiview MAE

MAEs (He et al., 2022) are a self-supervised learning approach designed to learn high-quality representations by reconstructing missing portions of the input. MAEs randomly mask a large fraction of the input data and train an encoder-decoder architecture to recover the masked content from the visible subset. In this work, we assume the encoder and decoder to be vision transformers (ViTs, Dosovitskiy et al., 2021).

$M(\cdot)$ applies a random mask to the input, where $\alpha$ is the masking ratio, and defines the number of patches removed from the input. We define the input as a sequence of patch tokens, where each token corresponds to a fixed-size image patch that is linearly embedded into a vector representation. We define the unmasked input tokens as $\boldsymbol{x}_{v_{\mathrm{vis}}}^{(i)} = M(\boldsymbol{x}_v^{(i)})$.

Let $\mathcal{T}_{\mathrm{vis}} \subseteq \{1, \dots, T\}$ denote the set of indices corresponding to the visible (unmasked) tokens after applying $M(\cdot)$. The number of unmasked tokens is therefore given by $T_{\mathrm{vis}} = |\mathcal{T}_{\mathrm{vis}}| = (1 - \alpha) \cdot T$, where $T$ is the total number of input tokens. The encoder $f_\theta$ processes only the unmasked input tokens $\boldsymbol{x}_{v_{\mathrm{vis}}}^{(i)}$ producing latent representations $\boldsymbol{z}_v^{(i)} = f_\theta(\boldsymbol{x}_{v_{\mathrm{vis}}}^{(i)})$. These are passed to a lightweight decoder $g_\phi$, which adds mask token placeholders and attempts to reconstruct the original input $\boldsymbol{x}_v$, including the masked parts. The model is trained by minimizing a reconstruction loss $\mathcal{L}_{\mathrm{Rec}}$ over only the masked positions:

$$\mathcal{L}_{\mathrm{Rec}}^{(i)} = \frac{1}{\alpha} \frac{1}{|\mathcal{V}^{(i)}|} \sum_{v \in \mathcal{V}^{(i)}} \sum_{t \notin \mathcal{T}_{\mathrm{vis}}} \left\| \boldsymbol{x}_{v_t}^{(i)} - \hat{\boldsymbol{x}}_{v_t}^{(i)} \right\|_2^2, \quad \text{where} \quad \hat{\boldsymbol{x}}_v^{(i)} = g_\phi(\boldsymbol{z}_v^{(i)}).$$

---

[1]The code is available at: `https://github.com/agostini335/MVMAE`.

Here, $\mathbf{x}_{v_t}^{(i)}$ denotes the original input view $v$ at masked position $t$, and $\hat{\boldsymbol{x}}_{v_t}^{(i)}$ the reconstruction of the same input position $t$. Please note that the reconstruction loss is only computed over masked input patches, but we feed only the visible or unmasked patches $\boldsymbol{x}_{v_{\mathrm{vis}}}^{(i)} = M(\boldsymbol{x}_v^{(i)})$. For more details, we refer to He et al. (2022).

In addition to the reconstruction loss, we introduce a study-level alignment term to encourage consistent representations across all views in the same study (Agostini et al., 2024; Sutter et al., 2024). Let $\mathcal{V}^{(i)} = \{v_1, \ldots, v_{n^{(i)}}\}$ denote the set of available views for study $i$, and let $\boldsymbol{z}_v^{(i)}$ be the encoder output (CLS and visible tokens only) for view $v$. We define the alignment objective as the average pairwise distance between encoder outputs for all ordered view pairs:

$$\mathcal{L}_{\mathrm{Align}}^{(i)} = \frac{1}{1-\alpha} \frac{1}{|\mathcal{P}^{(i)}|} \sum_{(u,v) \in \mathcal{P}^{(i)}} \sum_{t \in \mathcal{T}_{\mathrm{vis}}} d_{\mathrm{MSE}}\left(\boldsymbol{z}_{u_t}^{(i)}, \boldsymbol{z}_{v_t}^{(i)}\right), \tag{1}$$

where $\mathcal{P}^{(i)} = \{(u,v) \mid u,v \in \mathcal{V}^{(i)}, u \neq v\}$, and $d_{\mathrm{MSE}}(\cdot,\cdot)$ is the mean squared error (MSE). Importantly, alignment is computed only over visible encoder tokens; masked tokens are introduced later in the decoder.

The full objective for MVMAE over study $i$ is then:

$$\mathcal{L}^{(i)} = \mathcal{L}_{\mathrm{Rec}}^{(i)} + \beta \cdot \mathcal{L}_{\mathrm{Align}}^{(i)}, \tag{2}$$

where $\beta$ is the weighting between reconstruction and alignment loss. We additionally apply a beta annealing schedule, progressively increasing the alignment weight $\beta$ from zero to its target value over training to stabilize early-stage learning.

As introduced in the previous section, clinical studies may contain multiple radiographic projections, including repeated acquisitions of the same view type (e.g., two frontal views or a frontal–lateral–frontal sequence). To make the encoder aware of the projection type without assigning a separate head per view instance, we introduce a learnable modality embedding $\mathbf{e}_m \in \mathbb{R}^D$ for each projection category $m \in \{\text{frontal}, \text{lateral}, \text{unknown}\}$. Thus, all occurrences of a frontal view share the same embedding $\mathbf{e}_{\text{frontal}}$, even if multiple frontal images appear within the same study.

Let $\mathbf{x}_v^{(i)}$ denote the $v$-th view of study $i$ with associated modality index $m(v)$. After patchification, we add the regular positional encoding, and additionally shift every visible patch token by the corresponding modality embedding:

$$\mathbf{h}_t = \mathbf{p}_t^{\mathrm{pos}} + \mathbf{e}_{m(v)}, \quad t \in \mathcal{T}_{\mathrm{vis}}, \tag{3}$$

where $\mathbf{p}_t^{\mathrm{pos}}$ is the regular ViT positional embedding. The same $\mathbf{e}_{m(v)}$ is also added to decoder tokens, ensuring that reconstructions remain conditioned on the projection type. This design is analogous to segment embeddings in language models (Devlin et al., 2019), allowing the model to represent projection-specific acquisition bias while keeping a shared encoder across all views. In practice, we use $M = 3$ modality embeddings and learn the table $\{\mathbf{e}_m\}_{m=1}^M$ jointly with all other parameters.

## 4.2 MVMAE-V2T: Text as Additional Learning Signal

In addition to purely image-based pretraining, we investigate the use of radiology reports as an auxiliary supervision signal, rather than as an input modality, and we introduce MVMAE-Vision to Text (MVMAE-V2T). Each report is paired with its corresponding imaging study and serves only to guide visual representation learning during training. The text is never used as input at inference time or during downstream evaluation, ensuring a fair comparison with purely vision-based models. To incorporate this weak multimodal supervision, we adopt a captioning-style objective inspired by the CapPa framework (Tschannen et al., 2023). Specifically, for each view $\boldsymbol{x}_v^{(i)}$ of a study $i$ and its study report $\boldsymbol{r}^{(i)} = \left\{r_1^{(i)}, \ldots, r_{K^{(i)}}^{(i)}\right\}$ with $K^{(i)}$ tokens, we extend MVMAE with a transformer decoder $h_\psi$ that causally predicts each report token $r_k^{(i)}$ conditioned on all previous report tokens $r_{<k}^{(i)}$, as well as the encodings of the unmasked visual embeddings of the current

view $\boldsymbol{z}_v^{(i)}$ via cross-attention. To train $h_\psi$, we apply a cross-entropy loss in autoregressive fashion as follows:

$$\mathcal{L}_{CE}^{(i)} = \frac{1}{|\mathcal{V}^{(i)}|} \frac{1}{K^{(i)}} \sum_{v \in \mathcal{V}^{(i)}} \sum_{k=1}^{K^{(i)}} \ell_{ce}\left(\hat{\boldsymbol{r}}_{vk}^{(i)}; r_k^{(i)}\right), \tag{4}$$

where $\hat{\boldsymbol{r}}_{vk}^{(i)} = h_\psi\left(r_{<k}^{(i)}, \boldsymbol{z}_v^{(i)}\right) \in \mathbb{R}^{|\mathcal{W}|}$ are the logits of the $k$-th token given the previous tokens $r_{<k}^{(i)}$ and the unmasked view embeddings $\boldsymbol{z}_v^{(i)}$, $\mathcal{W}$ the token vocabulary, $r_0^{(i)}$ is a padding token and

$$\ell_{ce}(\boldsymbol{x}; y) = -\log \frac{\exp(x_y)}{\sum_{w \in \mathcal{W}} \exp(x_w)}. \tag{5}$$

We add this additional loss on top of the standard MVMAE loss and train MVMAE-V2T using

$$\mathcal{L}_{V2T}^{(i)} = \mathcal{L}_{\text{Rec}}^{(i)} + \beta \cdot \mathcal{L}_{\text{Align}}^{(i)} + \gamma \cdot \mathcal{L}_{CE}^{(i)}. \tag{6}$$

Where $\gamma$ is the weighting term for the cross-entropy loss. This joint optimization ensures that complementary information from multiple projections contributes coherently to the text generation pretext, reinforcing the study-level structure inherent in clinical data. Our approach leverages radiology reports solely as structured, domain-specific supervision to guide the visual encoder toward disease-relevant semantics, without relying on text for downstream reasoning or prediction.

Related vision–language pretraining methods such as SigLIP 2 (Tschannen et al., 2025) also employ captioning-style objectives, but rely on symmetric image–text alignment and additional mechanisms such as distillation and global–local consistency. In contrast, MVMAE-V2T treats text solely as an auxiliary supervision signal and deliberately avoids explicit image–text alignment, reflecting the clinical setting in which reports are narrative descriptions of images rather than an independent modality.

## 5 Experiments

Our aim is to investigate how multi-view pretraining with MVMAE influences downstream disease classification across different datasets and under varying constraints. These settings reflect realistic clinical conditions, where annotations are scarce, imaging protocols differ across institutions, and studies may contain variable numbers of views. We study four complementary questions:

(i) *How effective are the representations learned by a pretrained encoder when the model is fully finetuned on a downstream classification task?* In particular, we assess whether leveraging the multiview and multimodal data structure during pretraining leads to measurable improvements. We compare our approach against supervised baselines, structure-agnostic pretraining, and recent medical VLMs.

(ii) *How robust are MVMAE representations under limited supervision?* To simulate annotation scarcity, we finetune the pretrained encoder on progressively larger labeled subsets and compare against baselines, quantifying label efficiency.

(iii) *Does leveraging the structure at pretraining help with model calibration?* We evaluate whether multi-view pretraining leads to better-calibrated predictions compared to baselines.

(iv) *How does performance vary with the number of views per study?* We ablate the maximum number of views available during evaluation to test the robustness of multi-view pretraining when only a single view is provided at inference time, and the change when multiple views are available. To ensure a controlled comparison, we restrict this analysis to studies with exactly two available views.

Collectively, these experiments evaluate performance, label efficiency, and sensitivity to study structure.

### 5.1 Benchmark Models: From Unimodal Variants to Foundation Models

To contextualize our results, we benchmark MVMAE and MVMAE-V2T against a spectrum of pretrained encoders that represent different training regimes and levels of domain specialization. These baselines range

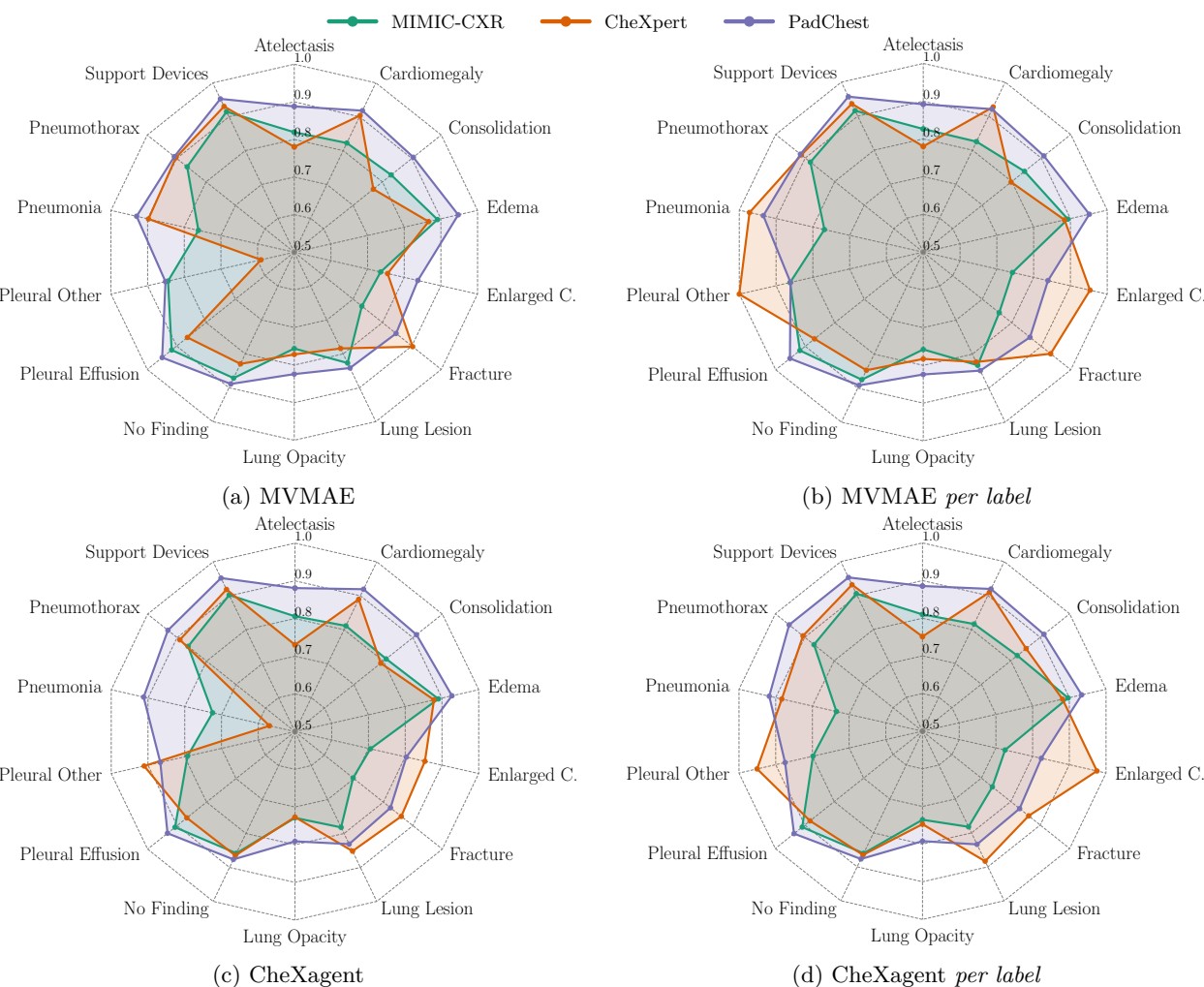

Figure 3: **Per-label AUROC comparison across datasets** (MIMIC-CXR, CheXpert Plus, PadChest). Left: overall performance with model selection done based on the best joint dataset. Right: model selection done to optimize per label results.

from variations of our own approach that are purely unimodal or do not enforce cross-view regularization, to state-of-the-art vision–language models trained on large-scale biomedical corpora or chest-X-ray–specific data. This progression allows us to test three hypotheses: (i) whether multi-view regularization is necessary beyond unimodal masked reconstruction, (ii) whether large but non–X-ray-specific biomedical pretraining confers an advantage, and (iii) how our approach compares to foundation models explicitly optimized for chest radiography interpretation. The evaluated benchmarks are described below.

**Supervised.** A fully supervised baseline that trains the backbone end-to-end on the multi-label classification task directly, without any pretraining, masking, or alignment. This represents the conventional approach used in many radiology benchmarks and serves as a control to assess how much benefit comes from our self-supervised pretraining. The same vision encoder architecture as in MVMAE is employed to ensure a fair and controlled comparison.

**Independent.** This ablation removes the cross-view alignment regularizer by setting $\beta = 0$. Each projection within a study is reconstructed independently, without explicitly encouraging shared latent representations between frontal, lateral, or unknown views. This method tests whether performance gains stem from the alignment mechanism itself, or simply from masked reconstruction. The same vision encoder architecture as in MVMAE is employed to ensure a fair and controlled comparison.

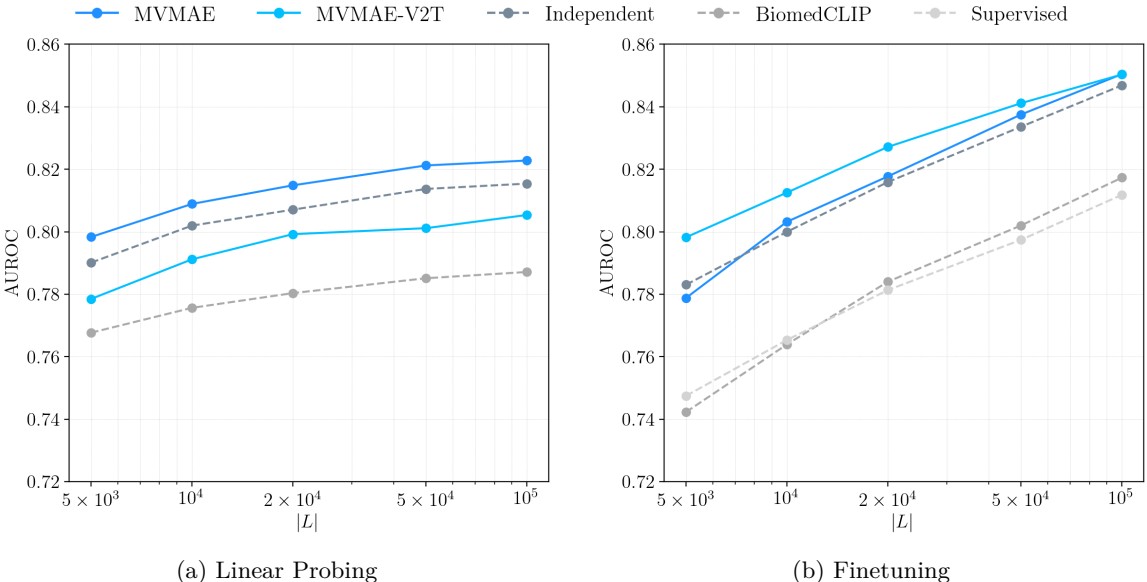

(a) Linear Probing

(b) Finetuning

Figure 4: **Label efficiency under finetuning**. Performance curves of the *Combined* dataset macro-average AUROC over 14 pathology labels, as a function of the number of labeled studies used for finetuning.

**BiomedCLIP.** Zhang et al. (2025) propose a large-scale biomedical VLM. It follows the CLIP (Radford et al., 2021) framework, jointly training a ViT-based image encoder and a PubMedBERT (Gu et al., 2021) text encoder on PMC-15M (Zhang et al., 2025), a dataset of over 15 million biomedical image–caption pairs mined from PubMed Central. This enables the model to generalize across a wide range of biomedical modalities, including radiology, pathology, microscopy, and illustrations. Compared to prior medical CLIP variants, it benefits from a much larger and diverse pretraining corpus, higher input resolution, and domain-specific adaptations.

**CheXagent.** Chen et al. (2024b) introduce a VLM for chest radiography. It is trained on CheXinstruct, a large-scale dataset of 8.5 million instruction–image–response triplets curated from 32 public sources, covering 35 distinct chest-X-ray interpretation tasks. The model combines a SigLIP-based image encoder with a decoder-only language model. This design allows it to perform a wide spectrum of tasks, including view classification, disease identification, temporal reasoning, VQA, phrase grounding, or radiology report generation. For fairness, we evaluate only the CheXagent vision encoder, not the full VLM, which itself was pretrained with self-supervised objectives before being integrated into the decoder-only architecture.

## 5.2 Implementation Details

For downstream classification, we use a late-fusion ensemble that averages prediction scores from unimodal single-view models for both the proposed MVMAE and baselines. A single multi-label classifier is trained to predict the fourteen diagnostic labels. The classifier is trained on the *Combined* dataset, obtained by merging studies from MIMIC-CXR, CheXpert, and PadChest. We report performance both on the *Combined* validation set and separately on each individual dataset. This setup enables a unified evaluation while still highlighting cross-institutional generalization. All models were trained with data augmentations common in self-supervised vision setups, including random resized cropping, horizontal flipping, and color jitter, to improve representation quality. Our MVMAE backbone is a ViT-Base (Dosovitskiy et al., 2021) with 12 layers, 12 attention heads, and 768-dimensional hidden embeddings, operating on $16 \times 16$ image patches. During pretraining, we apply a high masking ratio of 90%. The decoder is a lightweight transformer appended only during reconstruction. All experiments were conducted on an internal cluster using NVIDIA Grace Hopper Superchips with 40GB memory per GPU. Further implementation and trainig details are included in Appendix B.

## 5.3 Results

**Overall performance across datasets.** The first experiment evaluates the downstream classification performance of the proposed MVMAE compared to the baseline methods described in Section 5.1, when trained on the full labeled *Combined* dataset. The MVMAE encoder is finetuned end-to-end on the *Combined* data, and the same finetuning procedure is applied to BiomedCLIP and to the independent baseline to enable a direct comparison under identical conditions. In contrast, CheXagent is evaluated under a linear-probing setting, where only a classification head is trained while the pretrained encoder remains frozen, as its encoder has already been optimized for chest X-ray interpretation during large-scale pretraining. Model selection is performed using the macro-average AUROC over the fourteen labels on the *Combined* evaluation set. Results are reported in Tables 1 and 2, showing that MVMAE achieves the highest performance overall, with strong generalization across institutions and outperforming both supervised and VLM baselines.

Table 1: **Classification performance across datasets**. Comparison of MVMAE and baseline methods on the full labeled datasets. The table reports AUROC and F1 (mean ± std in three seeds) averaged over the fourteen labels per dataset, along with a combined metric averaging performance across all three. ViT-B/16 stands for ViT-Base architecture with 16 patch size, and ViT-L/14 for ViT-Large with 14 patch size.

| Model | Backbone | MIMIC-CXR | | CheXpert | | PadChest | | Combined | |
|---|---|---|---|---|---|---|---|---|---|
| | | AUROC | F1 | AUROC | F1 | AUROC | F1 | AUROC | F1 |
| Supervised | ViT-B/16 | 79.25 ± 0.42 | 25.30 ± 0.36 | 76.39 ± 0.95 | 23.77 ± 0.66 | 85.72 ± 0.28 | 27.16 ± 1.04 | 84.83 ± 0.25 | 29.33 ± 0.66 |
| Independent | ViT-B/16 | 81.28 ± 0.15 | 29.16 ± 1.19 | 79.65 ± 1.67 | 30.58 ± 3.05 | 88.17 ± 0.29 | 33.22 ± 0.22 | 86.60 ± 0.04 | 35.48 ± 0.18 |
| BiomedCLIP | ViT-B/16 | 78.88 ± 0.88 | 23.93 ± 2.29 | 76.72 ± 4.11 | 24.39 ± 1.87 | 84.87 ± 0.89 | 27.53 ± 3.51 | 83.97 ± 0.78 | 29.02 ± 3.36 |
| CheXagent | ViT-L/14 | 80.65 ± 0.13 | 28.88 ± 3.60 | 83.02 ± 0.22 | 33.54 ± 4.81 | 88.31 ± 0.05 | 28.39 ± 3.69 | 86.25 ± 0.04 | 33.56 ± 0.70 |
| MVMAE-V2T (ours) | ViT-B/16 | 81.20 ± 0.16 | 31.81 ± 6.12 | 80.87 ± 0.35 | 35.96 ± 9.35 | 88.27 ± 0.29 | 37.06 ± 6.74 | 86.74 ± 0.29 | 38.96 ± 6.59 |
| MVMAE (ours) | ViT-B/16 | **82.46** ± 0.26 | **33.70** ± 5.67 | **83.24** ± 0.81 | **37.56** ± 7.65 | **89.17** ± 0.21 | **39.69** ± 5.86 | **87.41** ± 0.03 | **41.27** ± 5.67 |

Table 2: **Top-5 classification performance across datasets**. Comparison of MVMAE and baseline methods on the full labeled datasets. The table reports Top-5 (labels) average AUROC and F1 (mean ± std over three seeds) per dataset, along with a combined metric averaging performance across all three.

| Model | Backbone | MIMIC-CXR | | CheXpert | | PadChest | | Combined | |
|---|---|---|---|---|---|---|---|---|---|
| | | AUROC | F1 | AUROC | F1 | AUROC | F1 | AUROC | F1 |
| Supervised | ViT-B/16 | 82.20 ± 3.18 | 38.84 ± 5.87 | 78.32 ± 6.03 | 45.47 ± 1.05 | 84.10 ± 5.47 | 29.45 ± 5.21 | 86.16 ± 3.14 | 38.09 ± 6.26 |
| Independent | ViT-B/16 | 86.27 ± 1.66 | 50.18 ± 3.68 | 83.49 ± 2.27 | 46.17 ± 4.82 | 89.55 ± 3.13 | 42.32 ± 3.29 | 88.99 ± 1.90 | 50.80 ± 3.61 |
| Chexagent | ViT-L/14 | 87.39 ± 0.98 | 52.28 ± 1.28 | **88.14** ± 0.52 | **55.30** ± 3.72 | 92.10 ± 0.50 | 40.74 ± 2.14 | 90.64 ± 0.48 | 53.37 ± 0.92 |
| MVMAE-V2T (ours) | ViT-B/16 | 89.10 ± 0.15 | **57.83** ± 1.61 | 86.76 ± 0.41 | 54.14 ± 1.63 | 91.53 ± 0.56 | 43.38 ± 0.34 | 91.65 ± 0.05 | 57.47 ± 0.94 |
| MVMAE (ours) | ViT-B/16 | 89.10 ± 0.08 | 55.01 ± 8.84 | 86.87 ± 1.48 | 52.34 ± 3.30 | **92.47** ± 0.95 | 40.41 ± 6.86 | **91.78** ± 0.10 | 55.32 ± 7.53 |

To better understand the distribution of performance across different conditions, Figure 3 visualizes the AUROC achieved per label in each dataset. The radar chart highlights substantial variation across disease categories and across datasets, reflecting both differences in label prevalence and the relative difficulty of certain findings (e.g., "Pneumonia" or "Pleural Other"). In our findings, labels corresponding to frequent and visually localized conditions yield higher scores, whereas labels for rare or textually ambiguous conditions remain challenging. We further observe that the relative difficulty of specific labels varies across datasets and models, suggesting that institutional biases, labeling practices, and acquisition protocols influence how well models generalize per pathology. In Figure 3, we show the MVMAE variant achieving the best overall classification performance across the *Combined* dataset, alongside the strongest baseline (CheXagent), and we include the remaining baselines in Appendix C. Notably, even when using CheXpert, where CheXagent performs better in terms of Top-5 AUROC, MVMAE achieves comparable or superior results for several individual labels, e.g., pneumonia, atelactasis, and fracture. To isolate label-wise performance, we also report results using per-label model selection, selecting the checkpoint that maximizes macro-average AUROC for each label independently (Figure 3, right). We note that part of the apparent per-label variation in CheXpert arises from the strong imbalance of its official validation split. Certain conditions (e.g., Pneumonia and Pleural Other) are represented by only a single positive study in this subset, which artificially lowers

AUROC values when using joint model selection. When model selection is instead performed on each label individually, as in Figure 3 *b* and *d*, these differences vanish, confirming that the observed gap is due to data scarcity rather than model limitations. Overall, the radar plots further reveal that MVMAE matches or exceeds CheXagent on difficult conditions.

**Label efficiency.** In Figure 11, we assess how well representations transfer under limited supervision and label availability. Models are finetuned on increasing subsets of labeled studies ($|L| \in \{5K, 10K, 20K, 50K, 100K\}$). This experiment evaluates how effectively each method benefits from increasing supervision, indicating how rapidly performance scales with available labels. Note that the labeled subsets are used only for downstream evaluation, i.e., linear probing or supervised finetuning, while all pretraining, including MVMAE, MVMAE-V2T, and baselines, remains strictly self-supervised with all samples. Because our focus is on label-efficient transfer, we restrict baselines to methods that did not use labels during pretraining, thereby excluding CheXagent. We benchmark two settings: full finetuning and linear probing. Under linear probing, MVMAE consistently outperforms all baselines, demonstrating more linearly separable representations and superior label efficiency. When finetuned, MVMAE-V2T further highlights the benefit of incorporating reports, especially in low-label regimes, although this advantage diminishes as more labeled data becomes available. In practical terms, achieving approximately 80% AUROC requires 5k labeled samples for MVMAE variants, 10k for the Independent baseline ($2\times$ more), and 50k for the Supervised baseline ($10\times$ more). These results indicate that MVMAE yields label-efficient representations, while MVMAE-V2T provides additional gains when finetuning under scarce label conditions.

Table 3: Brier score under linear probing on 20K samples on the same encoder-type models, lower is better.

| Model | MIMIC-CXR | CheXpert | PadChest | Combined |
|---|---|---|---|---|
| Supervised | **0.0885** $\pm$ 0.0005 | **0.0955** $\pm$ 0.0010 | **0.0718** $\pm$ 0.0003 | **0.0815** $\pm$ 0.0004 |
| Independent | 0.0915 $\pm$ 0.0005 | 0.0963 $\pm$ 0.0016 | 0.0750 $\pm$ 0.0005 | 0.0845 $\pm$ 0.0003 |
| BiomedCLIP | 0.0924 $\pm$ 0.0052 | 0.0980 $\pm$ 0.0061 | 0.0769 $\pm$ 0.0053 | 0.0859 $\pm$ 0.0053 |
| MVMAE-V2T (ours) | 0.0964 $\pm$ 0.0015 | 0.1017 $\pm$ 0.0014 | 0.0825 $\pm$ 0.0022 | 0.0905 $\pm$ 0.0018 |
| MVMAE (ours) | 0.0903 $\pm$ 0.0006 | 0.0974 $\pm$ 0.0008 | 0.0743 $\pm$ 0.0008 | 0.0835 $\pm$ 0.0007 |

**Calibration.** We evaluate probabilistic calibration using the Brier score, which measures the mean squared gap between predicted probabilities and outcomes, i.e., lower is better. As expected, the supervised classifier baseline reports the lowest Brier score, given its specialized setup with direct end-to-end classification. Under linear probing with 20K labeled samples and matched backbones, MVMAE reports the lowest Brier score among the pretraining baselines on MIMIC-CXR and PadChest, as well as the best combined result (see Table 3). Improvements over Independent and BiomedCLIP are modest but consistent, indicating that multi-view pretraining not only improves discrimination but also produces better calibrated confidence estimates across datasets.

**Effect of the number of views per study.** We further investigate how the number of available projections per study influences model performance. To ensure a fair comparison when assessing the effect of adding an additional view, we restrict the analysis to studies that contain exactly two views and finetune the models only on those. We compare the proposed MVMAE with its independent variant (trained without the cross-view alignment term) while varying the number of views included during evaluation, from single-view to two-views configurations. As shown in Figure 5, where we report the average $\Delta$AUROC over three random seeds, computed as the AUROC of MVMAE minus that of the independent; MVMAE consistently demonstrates a performance advantage over the independent variant across both settings. On the *Com-*

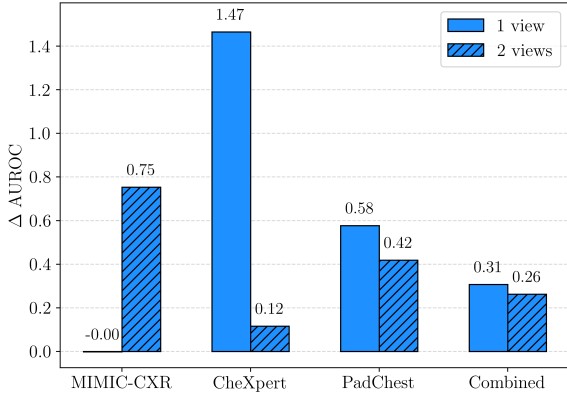

Figure 5: Average AUROC performance gain of MVMAE on Independent over three seeds.

*bined* dataset, the improvement is more pronounced under single-view evaluation, suggesting that multiview pretraining provides transferable benefits even when only one view is available. A similar trend is observed for CheXpert and PadChest, whereas for MIMIC-CXR, the improvement is noticeable only under two views.

Extended quantitative results from all our experiments, dataset-specific analyses, and additional visualizations are provided in Appendix C for completeness.

## 6    Discussion

Our findings highlight the importance of exploiting the inherent structure of medical imaging data for representation learning. Multi-view pretraining consistently improves downstream classification compared to training from scratch or using independent single-view objectives, confirming that structural coherence across projections provides a strong self-supervisory signal even without explicit labels or text.

Leveraging study-level structure during pretraining proves markedly more effective than enforcing consistency only at the supervision stage. Models that align views through soft regularization learn richer and more transferable representations than those combining predictions only at inference.

MVMAE also demonstrates superior label efficiency, achieving high performance with significantly fewer annotations, a crucial property in clinical domains where expert labeling is costly and often incomplete. Compared to large-scale vision–language models such as BiomedCLIP and CheXagent, MVMAE achieves competitive or better results without relying on text during pretraining.

A particularly interesting finding of our study is that MVMAE-V2T does not consistently outperform the vision-only MVMAE, despite leveraging additional textual supervision during pretraining. In several fine-tuning settings with full supervision, MVMAE without text matches or exceeds the performance of its V2T counterpart. This outcome is counterintuitive, as incorporating text is often assumed to improve representation quality. Our results suggest that when downstream labels are abundant, vision modality alone may already provide sufficient semantic structure, leaving limited room for auxiliary textual guidance and potentially introducing additional optimization complexity.

In contrast, MVMAE-V2T exhibits advantages in label-scarce regimes, which are highly representative of clinical practice. After full fine-tuning, it reliably outperforms MVMAE in these settings, suggesting that radiology reports function as a semantic bridge that structures the latent space around clinically meaningful concepts and improves transfer under limited supervision.

These findings indicate that MVMAE-V2T should not be interpreted as a universally superior alternative, but rather as an extension of MVMAE, optimized for data-scarce scenarios where textual supervision is available during pretraining.

We further find that MVMAE yields better-calibrated predictions among pretraining baselines, suggesting that cross-view regularization encourages more reliable confidence estimates alongside improved discrimination. Finally, training and evaluating across multiple datasets reveal that multi-view structural learning scales with data diversity and supports cross-institutional generalization.

Overall, these results highlight the importance of aligning pretraining objectives and evaluation protocols with the real-world organization of clinical data. Beyond chest X-rays, this framework—encompassing both MVMAE and MVMAE-V2T—naturally extends to temporal, multi-sequence, or multimodal studies, offering a scalable path toward clinically grounded foundation models built on structural and semantic supervision.

## 7    Conclusion

We introduced MVMAE, the first multi-view masked autoencoder that leverages the study-level structure of clinical imaging data to learn robust and transferable representations without relying on textual inputs during pretraining. By combining masked reconstruction with cross-view alignment, MVMAE effectively exploits the inherent multiview nature of radiology studies, yielding consistent improvements in performance and label efficiency across MIMIC-CXR, CheXpert, and PadChest. Building upon this foundation, we

further proposed MVMAE-V2T, which integrates radiology reports as an auxiliary text-based learning signal. This extension enhances semantic grounding and boosts performance in low-label regimes after full fine-tuning., while preserving fully vision-based inference. Together, MVMAE and MVMAE-V2T demonstrate that structural and textual supervision can act as complementary forces: structural alignment provides robustness and transferability, and text guidance enriches semantic understanding. Overall, these results establish a scalable framework for representation learning in medical imaging that leverages the inherent organization of clinical data. The same principles naturally extend to other structured modalities, such as temporal follow-ups or multi-sequence imaging, paving the way toward clinically grounded foundation models built on structural and semantic coherence rather than annotation scale. More broadly, the framework generalizes beyond medicine to any multimodal domain where structured relationships between views or modalities can serve as self-supervision for learning robust and transferable representations.

## Acknowledgments

This work was supported under project ID a135 as part of the Swiss AI Initiative, through a grant from the ETH Domain and computational resources provided by the Swiss National Supercomputing Centre (CSCS) under the Alps infrastructure, and by the LOOP Zurich as part of the application driver project supporting the LOOP Zurich Biomedical Informatics Platform (BMIP). TS and AA are supported by the grant #2021-911 of the Strategic Focal Area "Personalized Health and Related Technologies (PHRT)" of the ETH Domain (Swiss Federal Institutes of Technology). MV and SL are supported by the Swiss State Secretariat for Education, Research, and Innovation (SERI) under contract number MB22.00047. AR is supported by the StimuLoop grant #1-007811-002 and the Vontobel Foundation. SM acknowledges support from NSF, IARPA, HPI, CZI, Qualcomm, and Disney. ND and FN received research support from the Digitalization Initiative of the Zurich Higher Education Institutions (DIZH)- Rapid Action Call - under TRUST-RAD project. MK is supported by the UZH Global Strategy and Partnerships Funding Scheme and a Research Partnership Grant with China, Japan, South Korea and the ASEAN region (RPG 072023 18).

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

# A    Datasets and Study Structure

This section provides additional details on the four datasets used in our study: MIMIC-CXR, CheXpert, PadChest, and Chest X-ray. An illustration of the study-level structure of the combination of datasets is depicted in Figure 6, and an example of the study acquisition for one instance is illustrated in Figure 7.

A single study may contain multiple radiographic views, which we group into two macro-categories, *frontal* and *lateral*. Views for which this information is unavailable are categorized as *unknown*. All available views are retained during pretraining to avoid selection bias and to reflect the heterogeneity of clinical radiology data, where view metadata may be incomplete or inconsistently recorded. Moreover, retaining all views increases the effective training data volume, which we found to be beneficial despite the presence of potentially noisy signals. Views distribution for each dataset are described in Table 4

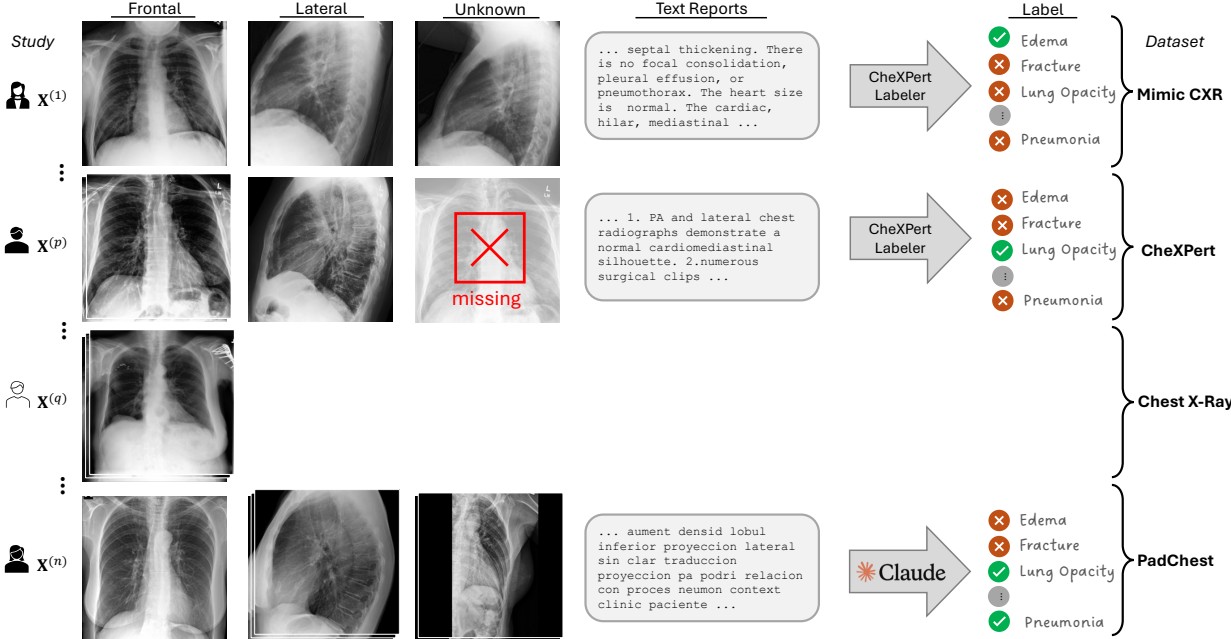

Figure 6: **Study-level structure representation**. Each row illustrates a representative study with three distinct views (frontal, lateral, and unknown), accompanied by corresponding report excerpts (text) and the final diagnostic labels (e.g., Edema, Lung Opacity, Fracture, Pneumonia) for the different datasets (MIMIC CXR, CheXPert, Chest X-Ray and PadChest). We treat each study as the fundamental unit of analysis, and various combinations of views are possible (e.g., including various frontal images (rows $p$ and $q$), or missingness (row $p$)). For Chest X-Ray dataset, no lateral or unknown views were available, nor text reports or labels.

Table 4: Distribution of radiographic views across datasets.

| Dataset | Frontal (F) | Lateral (L) | Unknown (U) |
|---|---|---|---|
| MIMIC-CXR | 239,931 | 116,555 | 15,465 |
| CheXpert+ | 191,071 | 32,391 | – |
| PadChest | 62,901 | 27,526 | 68,095 |
| ChestX-ray8 | 86,524 | – | – |

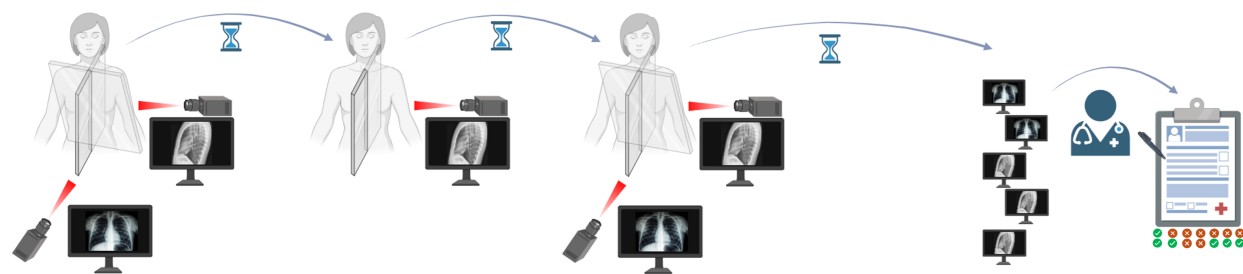

Figure 7: **Illustration of study acquisition**. For a single study, various views could be acquired at different frequencies. This figure illustrates an example of a study composed of three lateral view images and two frontal ones. With these, the radiologist writes a report from which labels are extracted.

Table 5 summarizes the key characteristics of each dataset, including the distribution of studies across the training, validation, and test splits, the total number of patients and images, and the cardinality of their original, unprocessed label sets.

Table 5: **Dataset Specifications**. A detailed breakdown of each dataset, including study counts for training, validation, and test splits, along with patient and image totals, label counts, and original image dimensions.

| | # Studies | | | # Patients | # Images | # Original Labels | Original Image Size |
|---|---|---|---|---|---|---|---|
| | **Train** | **Val** | **Test** | | | | |
| **MIMIC-CXR** | 222,758 | 1,808 | 3,269 | 65,086 | 371,951 | 14 | $2500 \times 3056$ |
| **CheXpert** | 187,511 | 200 | 500 | 64,725 | 223,462 | 14 | $320 \times 320$ |
| **PadChest** | 106,677 | 1,653 | 1,601 | 66,610 | 158,522 | 193 | $256 \times 256$ |
| **Chest X-ray** | 84,774 | 1,750 | 25,596 | 30,805 | 86,524 | 15 | $1024 \times 1024$ |

Figure 8 illustrates the label distribution in the training and validation splits for each dataset used in the fine-tuning phase (i.e., MIMIC-CXR, CheXpert, and PadChest), as well as for the *Combined* dataset. Note that in PadChest, the only available labels are 0 and 1 (indicating the absence or presence of a condition, respectively), whereas the other datasets include additional label values. This variation reflects the inherent diversity among the datasets; however, after preprocessing, all labels were binarized (0 or 1), ensuring a consistent and aligned representation across datasets. In Figure 9 we report the statistics after the binarization procedure. Finally, the following paragraphs describe the specific preprocessing steps applied to the labels to create a unified evaluation framework.

**Label Harmonization and Processing** To enable a unified analysis across all four datasets, we harmonized their diverse label sets into the 14 diagnostic categories defined by the CheXpert labeler: *Atelectasis, Cardiomegaly, Consolidation, Edema, Enlarged Cardiomediastinum, Fracture, Lung Lesion, Lung Opacity, No Finding, Pleural Effusion, Pleural Other, Pneumonia, Pneumothorax*, and *Support Devices*. The harmonization procedure was adapted to the characteristics and complexity of each dataset. For MIMIC-CXR and CheXpert, we directly utilized the binarized labels provided by the official CheXpert labeler, where explicit positive mentions were coded as 1, and all other states (negative, uncertain, or not mentioned) were coded as 0. Finally, the PadChest dataset required the most extensive harmonization due to its large and heterogeneous label set. It initially contained 433 labels, which were consolidated into 193 unique labels following text normalization (e.g., removal of extraneous spaces). In this case, we adopted Claude Code 2.0.35 to generate a mapping from the 193 PadChest labels to the 14 CheXpert categories using the following prompt *"I have these labels [list of 193 unique PadChest labels] and I have to map them to these other 14 labels: [list of 14 unique CheXpert labels], can you provide a csv file with the mapping?"*. This automated approach ensured complete and consistent coverage of all labels. The final mapping file is included in our repository for transparency and reproducibility.

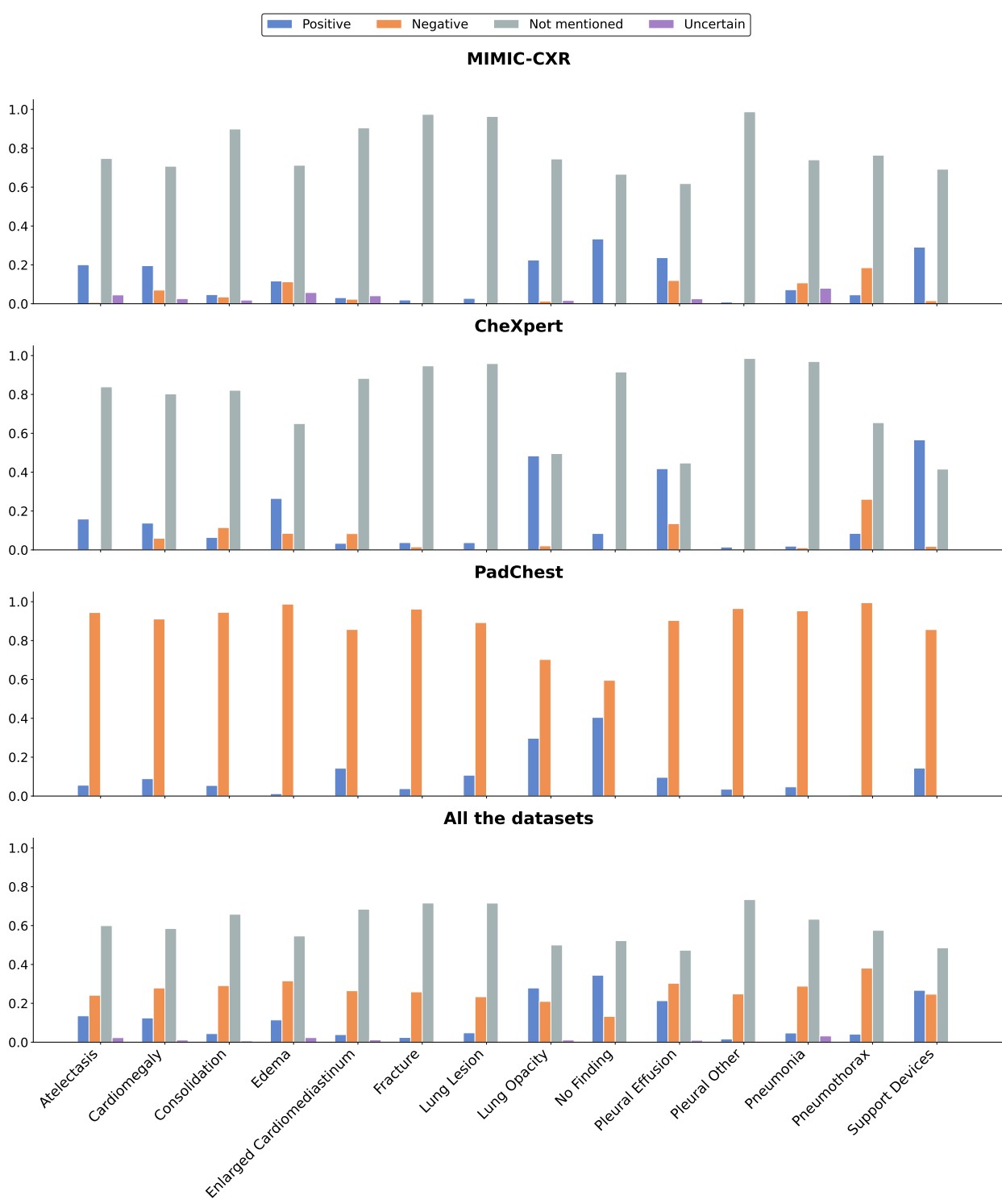

Figure 8: **Label Distribution.** Statistics for the MIMIC-CXR, CheXpert, and PadChest datasets, as well as for the *Combined* dataset in the training and validation splits. This figure illustrates how the frequency of each diagnostic label varies across individual datasets and the overall collection.

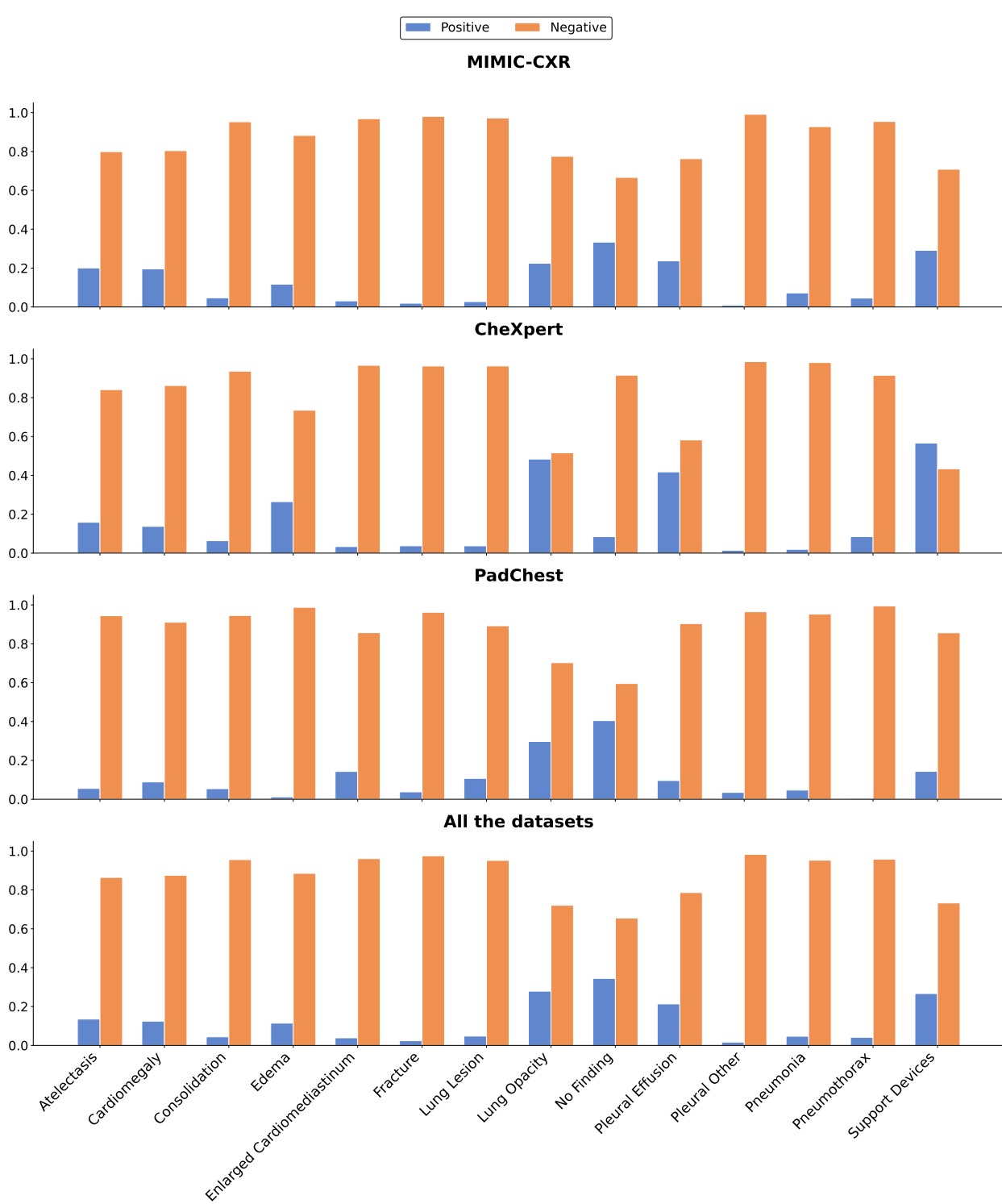

Figure 9: **Binary Label Distribution.** Statistics for the MIMIC-CXR, CheXpert, and PadChest datasets, as well as for the *Combined* dataset in the training and validation splits, after label binarization. This figure illustrates how the frequency of each diagnostic label varies across individual datasets and the overall collection.

# B    Implementation and Training Details

**Timing Analysis**

We compare the Independent baseline and MVMAE in terms of computational efficiency. Although MVMAE aligns multiple views, the additional cross view alignment objective incurs minimal cost relative to image encoding. We measure wall clock training time on identical HPC infrastructure, training both models for 500 epochs using 16 nodes. The Independent baseline requires *19,555* seconds, while MVMAE requires *19,601* seconds, indicating no meaningful difference in training time. These results show that MVMAE improves label efficiency without introducing a significant computational penalty. The optional vision to text objective introduces the expected overhead of text decoders and can be omitted when computational efficiency is a primary concern.

**Alignment Objective Implementation**

For cross-view alignment, we use a mean squared error (MSE) loss. This choice is motivated by its simplicity and effectiveness, enabling representation alignment with minimal computational cost. In preliminary experiments, we observe no substantial difference between MSE-based alignment and contrastive alignment using InfoNCE (Rusak et al., 2024). A more systematic exploration of alternative alignment objectives represents an interesting direction for future work.

**Training Parameters**

Pretraining of the MVMAE model reported in Tables 1 and 2 is performed for 1,200 epochs. We use a linear learning rate warmup for the first 5 epochs, followed by cosine annealing for the remaining training. The base learning rate is set to $lr = 2 \times 10^{-4}$ with a batch size of $bs = 64$. This learning rate schedule is adopted following prior work on masked autoencoders (Xiao et al., 2023).

The weighting between the reconstruction and alignment losses, denoted by $\beta$, follows a sigmoid annealing schedule, increasing from 0 to 1 with a steepness parameter of 0.02. This design choice aims to stabilize early-stage training by gradually introducing the alignment objective. For the MVMAE-V2T variant, we fix the weighting coefficient of the cross-entropy loss to $\gamma = 1$ as a default setting. Due to resource limitations, we did not conduct a sensitivity analysis of $\beta$ and $\gamma$, although we expect that further tuning of these hyperparameters could lead to additional performance improvements.

## C  Experiments

This section provides extended experimental details and complementary analyses to those presented in the main text. We include additional visualizations, dataset-specific label efficiency curves, and per-label classification tables to further examine how performance varies across pathologies, datasets, and model variants. These supplementary results offer a more granular view of the trends discussed in Section 5.3, reinforcing the main findings and illustrating the consistency and robustness of MVMAE and MVMAE-V2T across different evaluation settings.

**Overall performance across datasets: Results per label and dataset for all studied baselines**

To complement the aggregate results presented in Section 5.3, we report detailed per-label AUROC scores for each dataset and model variant in Table 6. These provide a fine-grained view of model behavior across the fourteen diagnostic categories on MIMIC-CXR, CheXpert, and PadChest, across studied baselines. These results allow for a closer inspection of pathology-specific trends and confirm that MVMAE consistently achieves competitive or superior performance compared to both unimodal and vision–language baselines. The accompanying radar plots in Figure 10 visualize these results across datasets, illustrating inter-dataset variability and highlighting how model selection based on specific pathologies or disease categories can yield targeted performance gains, an aspect particularly relevant for medical applications.

Table 6: Per-label classification AUROC on **MIMIC-CXR**. Results are shown for all compared models.

| Label | Supervised | Independent | BiomedCLIP | CheXagent | MVMAE (ours) |
|---|---|---|---|---|---|
| **Backbone** | ViT-B/16 | ViT-B/16 | ViT-B/16 | ViT-L/14 | ViT-B/16 |
| Atelectasis | 0.81 | 0.82 | 0.81 | 0.81 | 0.82 |
| Cardiomegaly | 0.80 | 0.81 | 0.80 | 0.81 | 0.82 |
| Consolidation | 0.81 | 0.82 | 0.80 | 0.81 | 0.83 |
| Edema | 0.88 | 0.89 | 0.88 | 0.89 | 0.89 |
| Enlarged C. | 0.68 | 0.73 | 0.74 | 0.71 | 0.74 |
| Fracture | 0.63 | 0.68 | 0.65 | 0.70 | 0.73 |
| Lung Lesion | 0.77 | 0.78 | 0.74 | 0.78 | 0.83 |
| Lung Opacity | 0.74 | 0.75 | 0.74 | 0.73 | 0.76 |
| No Finding | 0.86 | 0.86 | 0.85 | 0.86 | 0.87 |
| Pleural Effusion | 0.91 | 0.91 | 0.91 | 0.91 | 0.92 |
| Pleural Other | 0.74 | 0.83 | 0.80 | 0.79 | 0.84 |
| Pneumonia | 0.72 | 0.76 | 0.73 | 0.72 | 0.76 |
| Pneumothorax | 0.82 | 0.85 | 0.81 | 0.86 | 0.86 |
| Support Devices | 0.88 | 0.90 | 0.90 | 0.90 | 0.92 |

Table 7: Per-label classification AUROC on **CheXpert**. Results are shown for all compared models.

| Label | Supervised | Independent | BiomedCLIP | CheXagent | MVMAE (ours) |
|---|---|---|---|---|---|
| **Backbone** | ViT-B/16 | ViT-B/16 | ViT-B/16 | ViT-L/14 | ViT-B/16 |
| Atelectasis | 0.76 | 0.81 | 0.74 | 0.73 | 0.78 |
| Cardiomegaly | 0.91 | 0.92 | 0.94 | 0.89 | 0.90 |
| Consolidation | 0.76 | 0.75 | 0.82 | 0.79 | 0.77 |
| Edema | 0.86 | 0.86 | 0.83 | 0.88 | 0.87 |
| Enlarged C. | 0.88 | 0.86 | 0.87 | 0.85 | 0.75 |
| Fracture | 0.84 | 0.86 | 0.75 | 0.86 | 0.90 |
| Lung Lesion | 0.82 | 0.82 | 0.78 | 0.85 | 0.78 |
| Lung Opacity | 0.72 | 0.78 | 0.74 | 0.73 | 0.77 |
| No Finding | 0.80 | 0.82 | 0.84 | 0.86 | 0.83 |
| Pleural Effusion | 0.85 | 0.86 | 0.85 | 0.87 | 0.86 |
| Pleural Other | 0.66 | 0.73 | 0.70 | 0.91 | 0.59 |
| Pneumonia | 0.29 | 0.58 | 0.82 | 0.57 | 0.90 |
| Pneumothorax | 0.82 | 0.90 | 0.86 | 0.89 | 0.93 |
| Support Devices | 0.82 | 0.87 | 0.88 | 0.92 | 0.93 |

Table 8: Per-label classification AUROC on **PadChest**. Results are shown for all compared models.

| Label | Supervised | Independent | BiomedCLIP | CheXagent | MVMAE (ours) |
|---|---|---|---|---|---|
| **Backbone** | ViT-B/16 | ViT-B/16 | ViT-B/16 | ViT-L/14 | ViT-B/16 |
| Atelectasis | 0.80 | 0.85 | 0.83 | 0.88 | 0.89 |
| Cardiomegaly | 0.91 | 0.92 | 0.91 | 0.92 | 0.92 |
| Consolidation | 0.88 | 0.91 | 0.88 | 0.91 | 0.91 |
| Edema | 0.91 | 0.94 | 0.93 | 0.93 | 0.95 |
| Enlarged C. | 0.81 | 0.84 | 0.81 | 0.80 | 0.84 |
| Fracture | 0.79 | 0.84 | 0.78 | 0.82 | 0.85 |
| Lung Lesion | 0.78 | 0.82 | 0.79 | 0.83 | 0.84 |
| Lung Opacity | 0.80 | 0.82 | 0.80 | 0.79 | 0.82 |
| No Finding | 0.86 | 0.89 | 0.87 | 0.88 | 0.89 |
| Pleural Effusion | 0.94 | 0.95 | 0.94 | 0.93 | 0.95 |
| Pleural Other | 0.84 | 0.83 | 0.82 | 0.87 | 0.85 |
| Pneumonia | 0.89 | 0.91 | 0.82 | 0.91 | 0.93 |
| Pneumothorax | 0.86 | 0.95 | 0.82 | 0.93 | 0.91 |
| Support Devices | 0.91 | 0.95 | 0.80 | 0.95 | 0.95 |

**Label Efficiency per Dataset under Finetuning**

Figure 11 presents the dataset-specific label efficiency curves for MIMIC-CXR, CheXpert, and PadChest, respectively. These plots correspond to the same experiment described in Section 5.3, but show the trends separately for each dataset. For both linear probing and full finetuning, MVMAE and MVMAE-V2T demonstrate higher label efficiency compared to baselines, most significantly in the low-label regime, confirming the consistency of the results across datasets and highlighting the robustness of multi-view pretraining.

**Effect of Number of Views per Study: Detailed Results**

Table 9 expands on the analysis presented in **Effect of number of views per study** in Section 5.3 by reporting the absolute AUROC values across datasets. While the main text focuses on the relative performance gains ($\Delta$) when transitioning from single-view to dual-view evaluation, this table provides a more detailed view of the underlying scores. The results show that both models benefit from incorporating additional views; however, MVMAE with two views consistently achieves the highest performance.

Table 9: Comparison of MVMAE and Independent models under single- and dual-view evaluation using 5k-sample linear probing. Both models were evaluated only on studies containing two available views to ensure a consistent comparison. Results show the mean macro-average AUROC across three random seeds.

| Model | Setting | AUROC | | | |
|-------|---------|-----------|----------|----------|----------|
| | | MIMIC-CXR | CheXpert | PadChest | **Combined** |
| MVMAE | One view | 72.43 | 53.74 | 71.49 | 76.63 |
| | Two views (all) | **75.22** | **57.33** | **77.38** | **79.50** |
| Independent | One view | 72.43 | 52.28 | 70.92 | 76.32 |
| | Two views (all) | 74.46 | 57.21 | 76.96 | 79.23 |

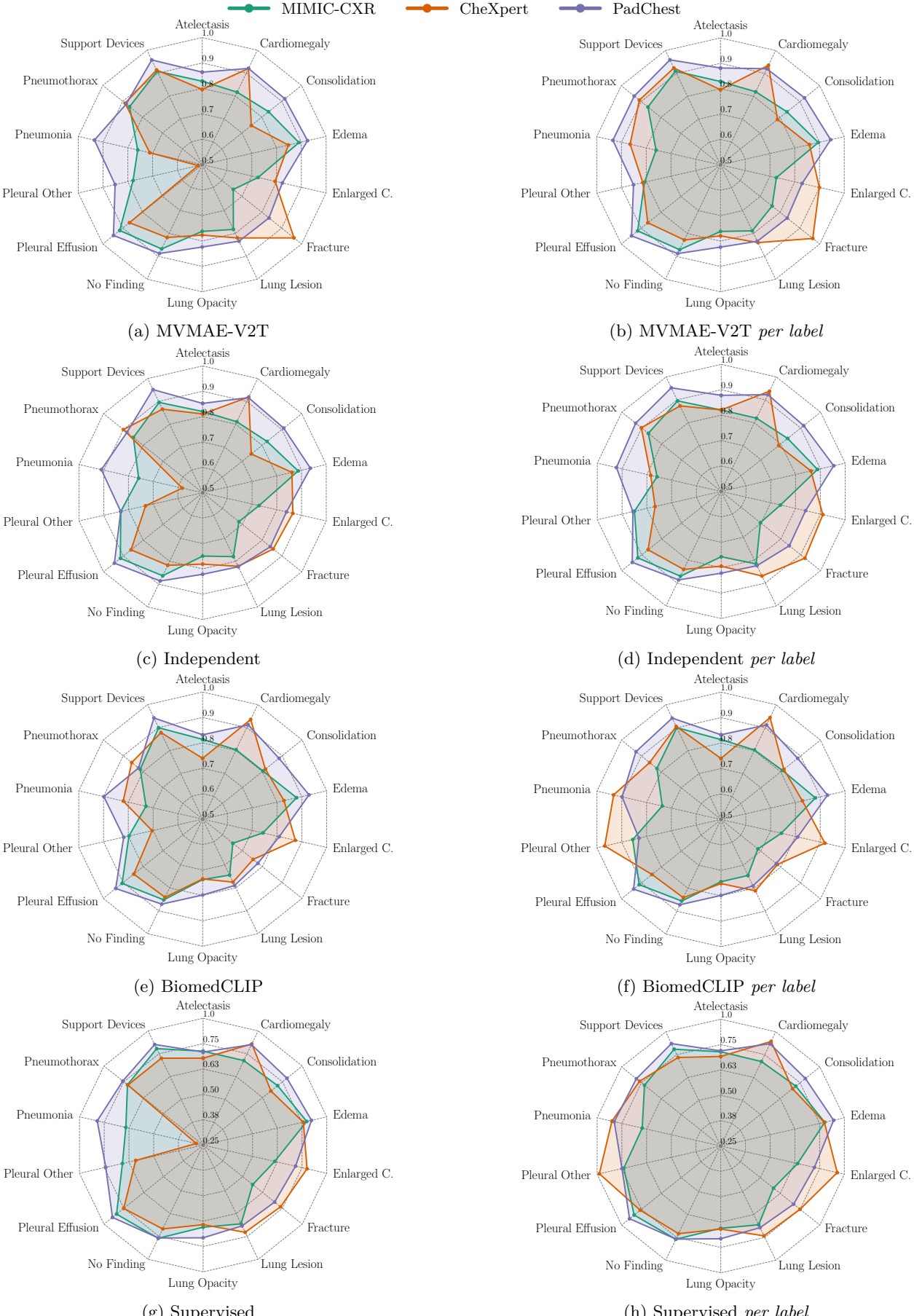

Figure 10: Per-label AUROC comparison across datasets for all models not shown in the main text.

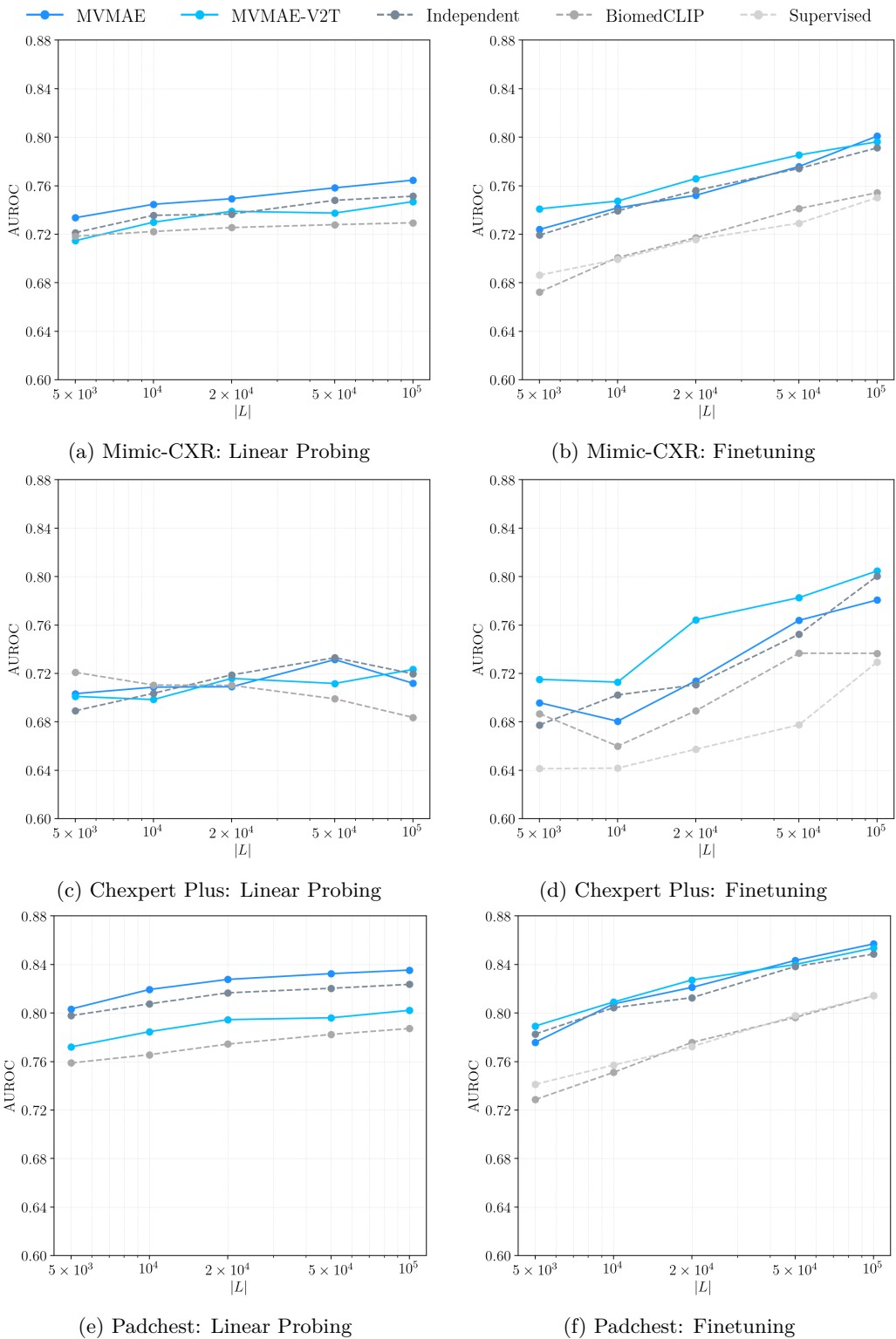

Figure 11: Label efficiency curves on Mimic-CXR, Chexpert Plus, and PadChest showing macro-AUROC across varying labeled data sizes.

