# OpenReview forum: "Structure is Supervision: Multiview Masked Autoencoders for Radiology"
_TMLR — Accepted by TMLR_

### Review · Reviewer_ZDv9 · 2025-12-22

**Summary Of Contributions:**

The paper proposes MVMAE, a study-centric self-supervised pretraining method for chest X-rays that leverages the natural multi-view structure of radiology studies. The method combines (1) masked image reconstruction and (2) a cross-view alignment loss to encourage view-invariant representations across multiple projections within a study. The paper also introduces MVMAE-V2T, which adds a vision-to-text objective using radiology reports during pretraining, while keeping inference vision-only. Experiments on MIMIC-CXR, CheXpert, and PadChest show MVMAE improves over supervised training and several baselines, while the V2T variant is presented as improving semantic grounding (especially in low-label regimes).

## STRENGTHS:

1. Radiology studies have inherent structure (multiple views + report), and exploiting that structure for SSL pretraining is well motivated.

2. Reconstruction + alignment is straightforward to implement and ablate, which has simple and clean objectives.

3. Evaluation tasks are broad, across multiple datasets and settings.

## Weaknesses:

1. MVMAE-V2T is not a symmetric image-text alignment objective (e.g., contrastive alignment like CLIP). Instead, it is a captioning objective: a transformer decoder predicts report tokens conditioned on visual embeddings via cross-attention, optimized with cross-entropy loss. This introduces an **inductive bias** toward report-predictive features, which could dilute fine-grained visual signals that matter for certain downstream tasks. This concern is consistent with Table 1: V2T is not consistently better than MVMAE in the full-label results (and MVMAE without text can be stronger on some metrics/datasets). The paper should more clearly justify why captioning is the best way to use text, and provide diagnostics showing when and why V2T helps versus hurts. At minimum, align the main claims with the mixed outcomes in the main table, and ideally provide variance/statistical significance.

2. The method introduces beta to balance reconstruction and alignment, and mentions a beta-annealing schedule. However, key details are unclear: what is the target beta value and the precise schedule, and how is the cross-entropy term weighted (it appears added with coefficient 1)? With multiple losses, results can be sensitive to scaling, and tuning can be expensive, especially across multi-dataset training. The paper would benefit from (i) explicitly reporting the chosen beta and schedule, (ii) optionally adding a gamma weight for the CE loss (even if gamma=1), and (iii) a small sensitivity analysis over beta (and gamma if used).

3. The paper states that reports are not used at inference to ensure fair comparison with purely vision-based models and to avoid relying on report availability or quality. This is reasonable for deployment settings where reports may be missing or noisy. However, it limits the scope: in many clinical workflows, text reports may be available at test time, and a multimodal inference model (image + text) could plausibly improve performance, robustness, or calibration. It would help to clarify whether the “vision-only inference” constraint is a core design goal or mostly a benchmarking choice. A small-scale experiment where reports are available during downstream training/inference (e.g., simple fusion or report-conditioned classifier) would clarify the trade-off.

**Audience:**

Yes

**Audience Explanation:**

This paper targets self-supervised pretraining for chest X-rays and introduces a simple, study-structure-driven objective (multi-view masked autoencoding + cross-view alignment) that shows consistent improvements across widely used medical imaging benchmarks. Even for readers outside radiology, the core idea—treating “structure” (multi-view/grouped observations) as supervision for representation learning—is broadly relevant and potentially transferable to other domains with grouped or multi-sensor data. The mixed outcome of the vision-to-text extension is also informative for the community, as it clarifies when adding language supervision may or may not help.

**Claims And Evidence:**

No

**Claims Explanation:**

I am not fully convinced that we should incorporate text into learning to provide better downstream performance. At least, the current design seems to be suboptimal. However, I could misunderstand the performance reported in the table. Please correct me.

**Requested Changes:**

1. Explain more explicitly that V2T is a captioning-style loss (vision-conditioned report generation), not a symmetric image–text alignment objective. Discuss what representation properties this is expected to encourage, and why this inductive bias should help downstream classification/transfer.

2. Table 1 suggests MVMAE-V2T is not consistently better than MVMAE (vision-only). Please (i) align the narrative/abstract with these outcomes, (ii) report mean±std over multiple seeds, and (iii) indicate which improvements are statistically significant.

3. Specify the exact beta value(s) used for the alignment loss, the annealing schedule, and how the cross-entropy term is weighted (if coefficient=1, justify). Add a small sensitivity study over beta (and gamma for CE if applicable) to show robustness and reduce concerns about expensive tuning.

---

> ### Author Response · Authors · 2026-01-23
> **Answer to reviewer ZDv9 [1/2]**
>
> >  Strengths
>
>
> We thank the reviewer for their constructive feedback and for recognizing that our 'study-centric' approach is well-motivated, straightforward to implement, and rigorously benchmarked.
>
>
>
> > 1. MVMAE-V2T vs MVMAE
>
>
> We clarify that the motivation for using a generative (captioning) objective rather than a symmetric contrastive alignment (such as CLIP) was to more faithfully model the data generation process in clinical radiology. In practice, radiology reports are narrative descriptions conditioned on visual evidence; therefore, a captioning-style loss encourages the model to learn semantic grounding in a way that mirrors clinical workflows. While symmetric contrastive models (e.g., BiomedCLIP) are included in our baselines, our results in Table 1 show that they do not consistently outperform our proposed architecture. Crucially, we believe the potential risk of "diluting" visual signals is mitigated by our joint objective: the MAE reconstruction loss ($L_{rec}$) operates in parallel with the captioning loss, ensuring the encoder retains the high-fidelity, fine-grained visual details necessary for reconstruction even as it learns high-level semantic features.
> We do not claim that the captioning objective or adding the text supervision (MVMAE-V2T) is universally superior to the base MVMAE model; rather, it represents a specialized variant that is particularly effective in low-label regimes, where the auxiliary textual signal provides a necessary semantic bridge. We acknowledge that in full-data settings, the inductive bias of text may lead to the mixed outcomes observed in the main table. To address this, we are updating the manuscript to discuss the differences in performance of the MVMAE-V2T variant and will include statistical significance (mean ± std) across all tables to provide a more rigorous basis for these comparisons.
>
>
> > 2. Training schedule and optimization parameters
>
> We will detail the hyperparameters and training schedules in the updated manuscript to address the concerns regarding reproducibility. Specifically, for the MVMAE model in Table 1, the cross-view alignment weight $\beta$ follows a sigmoid annealing schedule, increasing from 0 to 1 with a steepness parameter of 0.02. This annealing phase begins immediately following the initial learning rate warmup. Regarding the learning rate itself, we use a linear warmup for 5 epochs, followed by a cosine annealing schedule for the remainder of the 1195 epochs pretraining phase. We motivate this scheduler based on prior work on MAE and self-supervised learning paradigms [1, 2].
> Regarding the vision-to-text objective, the reviewer is correct that the cross-entropy term was weighted with a coefficient of 1. We will reformulate the objective in the revised text to include a $\gamma$ parameter. While we maintained $\gamma=1$ for our experiments, incorporating this notation provides a more flexible framework for future research to better control the balance between textual and visual supervision.
> Finally, we would like to clarify that we did not perform extensive hyperparameter tuning for $\beta$ or $\gamma$ in the loss weighting; the values were chosen as reasonable defaults. While we acknowledge that a sensitivity analysis would be beneficial, it is currently computationally unfeasible given the massive scale of the multi-dataset pretraining (comprising close to 1 million images). However, the fact that MVMAE achieves state-of-the-art results without exhaustive tuning suggests that the framework is robust and that further performance gains could potentially be realized through optimization of these parameters, though such optimization was not the primary focus of this work.
>
>
> [1] Xiao, J., Bai, Y., Yuille, A., & Zhou, Z. (2023). Delving into masked autoencoders for multi-label thorax disease classification. In Proceedings of the IEEE/CVF Winter Conference on Applications of Computer Vision (pp. 3588-3600).
>
>
> [2]He, K., Chen, X., Xie, S., Li, Y., Dollár, P., & Girshick, R. (2022). Masked autoencoders are scalable vision learners. In Proceedings of the IEEE/CVF conference on computer vision and pattern recognition (pp. 16000-16009).

---

> > ### Author Response · Authors · 2026-01-23
> > **Answer to reviewer ZDv9 [2/2]**
> >
> > > 3. Leveraging text and report availability
> >
> >
> > We clarify that the "vision-only inference" constraint is a core design goal rooted in the clinical reality of chest X-ray (CXR) workflows, rather than a mere benchmarking choice. In the specific context of radiology, reports are typically a narrative function of the visual evidence. If a radiology report is already available at inference time, the diagnostic task is essentially complete; in such cases, one would simply use a text-labeler or NLP tool to extract findings (labels) rather than deploying an additional model to "re-discover" what is already documented. Our motivation is to provide the best possible diagnostic tool for scenarios where the report does not yet exist or where a second visual opinion is required. We emphasize that this focus is specific to CXR and is not intended as a generalization for other medical domains. In other areas of healthcare, such as analyzing ECGs alongside patient history, or incorporating longitudinal medical notes, textual data provides complementary information that is not present in the image. In those settings, a multimodal ensemble (image + text) would indeed be the superior choice at inference time. For the current scope of this work, however, we treat reports as a form of "expert supervision" during training to sharpen the visual encoder's representations, which can then be deployed in real-world settings where only the images are available.
> >
> >
> > > Concern on claims and evidence to support the submission
> >
> >
> > We thank the reviewer for raising this concern. We have addressed this in our first response above, where we clarify that the vision-to-text (V2T) objective is intended as a specialized auxiliary signal rather than a universal replacement for pure visual pretraining. As discussed, while the captioning-style loss may not lead to consistent gains across all high-data regimes, it provides significant benefits in low-label scenarios (when fully fine-tuned) by bridging visual features with high-level clinical semantics. We believe this is relevant to the community, as the reviewer pointed out, as it opens new grounds for studying the contribution of text in the pretraining. We have revised our claims in the manuscript to more accurately reflect these mixed outcomes and have added statistical significance to the tables to provide a clearer, more transparent view of where and when the text-based supervision provides a meaningful advantage.
> >
> > > Final summary
> >
> >
> > Thanks again for the summary. We refer the reviewer to the inline answers for each of the requested changes.

---

### Review · Reviewer_czM9 · 2026-01-11

**Summary Of Contributions:**

**Summary**

This paper introduces MVMAE, a self-supervised pretraining framework for chest radiography that exploits the inherent multi-view structure of radiology studies. MVMAE combines masked image reconstruction with cross-view alignment. The authors extend this with MVMAE-V2T, which incorporates radiology reports as auxiliary signals during pretraining while maintaining vision-only inference. Experiments across MIMIC-CXR, CheXpert, and PadChest demonstrate consistent improvements over supervised and vision-language baselines, particularly in low-label regimes.

**Strengths**
1. Well Written: The paper is written in a good manner making it easy to follow,
2. Clinically motivated: The chosen problem statement and approach of leveraging the structured nature of radiology studies with multiple projections, is relevant and well motivated.
3. Comprehensive Evaluation: The experimental design is good. They evaluate their methods on 3 datasets, and analyze the per-label behavior of their models.
4. Strong Baselines: The authors compare their models, to CheXagent (VLM) and BiomedCLIP.

**Weakness**
1. MVMAE-V2T under performs MVMAE nearly across all comparisons. I feel authors should address this surprising result in the main text. What could be the reasons for this? Does text introduce noise?
2. Statistical Significance of Tables-1,2,3 : Were the results shown in Tables 1,2 averaged over random seeds, if so please report them in the main text. If not, I would recommend the authors to report the mean performance and std. dev instead of just performance.
3. Brier Scores : The values in table 3 seem very closeby (with the differences being < 0.01). It would be good if authors could do a significance testing to support these results.
4. Is it a fair comparison to use ViT-B/16 to compare against CheXagent (which uses ViT-L/14) , as it has 3x more parameters?

**Audience:**

Yes

**Audience Explanation:**

Yes, this work would be relevant to the medical AI community and is an incremental step towards improving AI's performance on medical topics.

**Claims And Evidence:**

Yes

**Claims Explanation:**

The paper presents convincing, accurate, clear evidence in forms of the tabular results and result plots. The experimental design makes sense and is well thought of.

**Requested Changes:**

The following additions would strengthen your work:

- Statistical Significance of the results (Table 1,2,3)
- Address why MVMAE-V2T under performs as compared to MVMAE in main text.

---

> ### Author Response · Authors · 2026-01-23
> **Answer to reviewer czM9**
>
> > Strengths
>
>
> We thank the reviewer for their positive assessment of our work. We are pleased that they found the paper well-written and easy to follow, and we appreciate the recognition of our clinical motivation and the comprehensiveness of our evaluation against strong baselines.
>
>
> > 1. MVMAE-V2T vs MVMAE
>
> We clarify that the motivation for incorporating the vision-to-text (V2T) component is to reflect the clinical reality in which radiology reports are narrative descriptions derived from visual evidence. As noted by the reviewer, MVMAE-V2T does not consistently outperform the vision-only MVMAE in full-data regimes. However, we found it noteworthy that, after full fine-tuning in label-scarce settings (e.g., the 5k–20k sample regime in Figure 4), where the impact of self-supervised pretraining is most critical, the V2T variant consistently achieves better downstream performance. In these scenarios, the textual supervision acts as a "semantic bridge" that helps the model organize the latent space more effectively than vision alone. We believe this regime-dependent behavior is an informative result for the community. We have addressed this in further detail in our response to Reviewer ZDv9, and we will update the manuscript to more clearly characterize MVMAE-V2T as a specialized variant for data-scarce scenarios, highlighting its performance differences relative to the vision-only version.
>
> > 2. Statistical significance of tables
>
> We are updating all tables in the revised manuscript to report the mean performance along with the standard deviation across three random seeds. We agree that providing these measures of variance is essential for a rigorous assessment of the performance gains and to confirm the stability of our findings across different initialization and sampling conditions.
>
>
> > 3. Brier scores
>
>
> As in question 2, we are updating the Brier scores results in the revised manuscript to report the mean performance along with the standard deviation across three random seeds.
>
>
> > 4. ViT Large and ViT Base architecture in baselines
>
> From our point of view, this comparison actually highlights the efficiency of our approach. We deliberately chose the ViT-B/16 architecture for MVMAE to demonstrate that by leveraging clinical structure as supervision, a smaller model can achieve or exceed the performance of much larger models. While CheXagent utilizes a ViT-L/14, which contains roughly 3x more parameters, this was a design choice by its authors, presumably because the larger capacity was necessary for their specific training objectives. By outperforming them with a ViT-B backbone, we show that our "study-centric" pretraining is highly parameter-efficient. Furthermore, using a smaller model is a practical necessity for many medical institutions that lack the massive computational resources required to train or deploy ViT-L/14. Being able to reach state-of-the-art results with a more compact architecture is a core strength of our method, rather than a limitation of the comparison.
>
>
> > Final summary
>
>
> Thanks again for the summary. We refer the reviewer to the inline answers for each of the requested changes.

---

### Review · Reviewer_DcLe · 2026-01-19

**Summary Of Contributions:**

The paper introduces Multiview Masked Autoencoder (MVMAE), a self-supervised pretraining framework designed specifically for the structured nature of radiology studies. Rather than treating radiographs as independent images, the authors propose a "study-centric" approach that leverages the natural redundancy found in multiple projections (e.g., frontal and lateral views) of the same patient encounter.

The methodology combines two primary objectives:

- Masked Image Reconstruction: Capturing local visual details by recovering masked patches within individual views.



- Cross-View Alignment: Encouraging view-invariant representations by regularizing embeddings across different projections of the same study.


Additionally, the authors extend this into MVMAE-V2T, which incorporates an auxiliary vision-to-text objective. This uses radiology reports during pretraining to enhance semantic grounding while ensuring the model remains a vision-only encoder during inference, avoiding dependence on text at test time.


# Key Strengths

- Structural Alignment with Clinical Reality: The method exploits the inherent organization of clinical data (multiple views per study), which is a powerful yet often underutilized signal in medical AI.




- High Label Efficiency: The framework demonstrates significant gains in low-label regimes. For instance, it achieves approximately 80% AUROC with only 5,000 labeled samples, whereas a supervised baseline requires 50,000 samples to reach a similar performance level.




- Robust Benchmarking: The authors evaluate their model across three large-scale public datasets (MIMIC-CXR, CheXpert, and PadChest), demonstrating consistent performance and cross-institutional generalization.



- Improved Calibration: Beyond simple classification accuracy, MVMAE yields better-calibrated confidence estimates (measured by Brier score) compared to independent or vision-language baselines.

# Key Weaknesses

- Incremental Technical Novelty: While the application is highly specialized and effective, the underlying technical components are largely adapted from existing methodologies. The novelty lies in the synthesis and domain adaptation—combining standard Masked Autoencoders (MAE) with standard alignment techniques and captioning objectives (CapPa) —rather than the invention of new architectural primitives. Overall, the introduced framework aligns with existing wokrs quiet a lot, such as SigLIP2 etc.

**Tschannen M, Gritsenko A, Wang X, et al. Siglip 2: Multilingual vision-language encoders with improved semantic understanding, localization, and dense features[J]. arXiv preprint arXiv:2502.14786, 2025.**

- Dependency on Pre-existing Labelers: For dataset harmonization, the work relies heavily on the CheXpert labeler and automated tools (Claude) to map disparate label sets (e.g., mapping PadChest’s 193 labels to CheXpert’s 14). Any inherent bias or noise in these labeling tools could propagate into the evaluation metrics.



- Computational Overhead: Jointly processing multiple views with high masking ratios (90%) and additional text decoders increases the complexity of the pretraining phase compared to simple unimodal SSL approaches.

**Additional Comments:**

While the proposed method is undeniably effective and highly aligned with the specific nature of radiological data, a critical reviewer would note that the **intrinsic technical novelty is constrained**, as it primarily re-assembles existing SOTA components:

## The framework is a combination of three well-established paradigms:

1.
**Masked Autoencoders (MAE)**: The reconstruction objective () and the high masking ratio (90%) are directly adopted from the original MAE work.


2.
**Multiview Alignment**: Using MSE or contrastive-like losses to align different views of the same object is a standard technique in multiview SSL (e.g., SimCLR/MoCo adaptations).


3.
**CapPa/Image Captioning**: The V2T extension uses a standard transformer decoder to causally predict report tokens, a method already explored in frameworks like CapPa.



## Domain-Driven Adaptation over Algorithmic Innovation

The "novelty" here lies not in a new mathematical operator or a novel transformer architecture, but in the **domain-specific engineering** of the objective functions:

*
**Modality Embeddings**: The use of learnable embeddings () for frontal/lateral views is analogous to BERT’s segment embeddings.


*
**Beta Annealing**: The schedule for weighting reconstruction vs. alignment is a common optimization heuristic in VAEs and multimodal learning.

**Audience:**

Yes

**Audience Explanation:**

While the technical components are familiar, the application of structural constraints to medical foundation models is of high interest to researchers in medical AI and self-supervised learning. The findings regarding how multiview alignment compensates for limited labels provide a valuable blueprint for other structured modalities like MRI or CT.

**Broader Impact Concerns:**

No, The paper uses de-identified public datasets and aims to improve the robustness of medical diagnosis.

**Claims And Evidence:**

Yes

**Claims Explanation:**

The evidence provided is robust and directly addresses the paper's claims:


Empirical Gains: MVMAE consistently outperforms supervised baselines and domain-specific VLMs like BiomedCLIP.



Label Efficiency: The model demonstrates a "10x gain" in low-label regimes; achieving 80% AUROC with 5k samples compared to the 50k samples required by a supervised baseline.


Calibration: The use of the Brier score proves that multi-view pretraining leads to better-calibrated confidence estimates, which is critical for clinical reliability.

**Requested Changes:**

The following adjustments are essential to support the paper's claims regarding the necessity and robustness of the specific multiview framework:

*
**Quantitative Justification of latent MSE vs. Contrastive Objectives**: The paper uses a MSE loss for . Since contrastive losses (e.g., InfoNCE) are more standard in multiview SSL, the authors must provide a brief comparative analysis or stronger theoretical justification for why a simple distance-based alignment is sufficient for radiographs.


*
**Ablation on "Unknown" View Noise**: The framework incorporates "Unknown" views into the training pipeline. It is critical to demonstrate whether these undefined views act as a regularizer or introduce noise that could degrade the alignment of high-quality Frontal-Lateral pairs.


*
**Clarification of Label Mapping Accuracy**: The use of an LLM (Claude) to map 193 PadChest labels to 14 CheXpert categories is a practical choice but introduces potential systematic error. A small-scale validation by a human expert on this mapping is necessary to ensure the "Combined" dataset metrics are not artifactually inflated by incorrect label grouping.



### Strengthening Suggestions

The following changes would improve the depth and transparency of the work but are not strictly required for acceptance:

*
**Visual Interpretation of Cross-View Information**: While the paper proves the method works via AUROC , providing attention maps (e.g., Grad-CAM) would strengthen the work by showing if  forces the model to focus on anatomically consistent regions across views (e.g., the same nodule in both frontal and lateral projections).


*
**In-depth Complexity Analysis**: Given that the novelty is primarily in the assembly of existing tools (MAE + Alignment + Captioning) , a more detailed discussion on the training wall-clock time and VRAM overhead compared to the "Independent" baseline would help practitioners assess the cost-benefit ratio of the structural supervision.

*
**Expanded Discussion on Technical Modesty**: The authors should more explicitly acknowledge that the primary contribution is the "domain-specific synthesis" of existing SSL primitives rather than the invention of new architectural components. This transparency aligns better with the TMLR criterion regarding modest but clear contributions.

---

> ### Author Response · Authors · 2026-01-23
> **Answer to reviewer DcLe [1/3]**
>
> > Strengths
>
>
> We thank the reviewer for their thorough assessment of our work. We are particularly encouraged by their recognition of our 'study-centric' approach as a powerful and clinically aligned signal, and we will answer point-by-point to the concerns in line.
>
>
> > Incremental Novelty and SigLIP 2
>
>
> We agree that our contribution lies in the domain-specific synthesis and adaptation of existing methods rather than proposing a new architecture. However, we argue that the primary novelty of this work is the introduction of a structure-based supervision paradigm that has not been previously explored in this manner. Our goal was to demonstrate that the inherent clinical organization of data (multiple views of one study) can be formalized into a powerful, yet simple, self-supervisory signal.
>
> Regarding the comparison to SigLIP 2, we thank the reviewer for this pointer; it is indeed a relevant work that we are including and discussing in our updated manuscript. While SigLIP 2 also utilizes a captioning loss (similar to our V2T extension and other  works like CapPa), its framework is significantly more complex, involving symmetric image-text alignment, distillation, and global-local consistency objectives. Our approach differs in three fundamental ways: (i) Modeling Clinical Reality: Unlike SigLIP 2, we deliberately avoid an alignment loss between image and text. As mentioned in our response to Reviewer ZDv9, we focus on the reality of CXR data where the report is a narrative function (caption) of the image, rather than a separate entity to be "aligned." (ii) Fine-Grained Visual Preservation: We incorporate a pixel-level MAE reconstruction loss ($L_{rec}$), which ensures the model retains fine-grained visual features that are often lost in pure vision-language models like SigLIP or CLIP. (iii) Structural vs. Multi-Modal: Our core innovation is the cross-view alignment of different images within the same study, a "structural" signal that is distinct from the image-text focus of the SigLIP family. We believe our method is more fitting for the specific constraints and data generation processes of the medical reality, and its simplicity is a strength that makes it more accessible for clinical deployment. We will use the discussion of SigLIP 2 to better highlight these distinctions in the revised related work section.
>
> > Labelers can introduce biases
>
>
> We agree with the reviewer that relying on automated labelers is a limitation; however, it is the established standard for large-scale radiology research. The ground-truth labels for the most widely used benchmarks, such as MIMIC-CXR and CheXpert, were themselves generated using the CheXpert automated labeler [1]. By applying the same logic to map PadChest's 193 labels to the 14 CheXpert categories, we maintain internal consistency across our "Combined" dataset. Furthermore, obtaining expert human annotations for millions of images is practically unfeasible. Since all major baselines (e.g., CheXagent, BiomedCLIP) rely on these same automated tools, our benchmarking remains fair and aligned with current state-of-the-art practices. We will add a discussion on this to our limitations section.
>
> [1] Irvin, J., Rajpurkar, P., Ko, M., Yu, Y., Ciurea-Ilcus, S., Chute, C., ... & Ng, A. Y. (2019, July). Chexpert: A large chest radiograph dataset with uncertainty labels and expert comparison. In Proceedings of the AAAI conference on artificial intelligence (Vol. 33, No. 01, pp. 590-597).

---

> > ### Author Response · Authors · 2026-01-23
> > **Answer to reviewer DcLe [2/3]**
> >
> > > Computational analysis
> >
> >
> > To address the concern regarding computational complexity, we clarify that the overhead of our framework is minimal compared to standard approaches. While we process multiple views, the high masking ratio (90%) significantly reduces the number of patches processed by the transformer blocks, leading to better computational efficiency. The only additional operation is the cross-view alignment objective, which is computationally negligible relative to the encoding of image patches. To demonstrate this empirically, we compared the wall-clock time for pretraining the "Independent" baseline (the same architecture without structural supervision) against MVMAE on the same HPC infrastructure. Both models were trained for 500 epochs using 16 nodes: The Independent baseline takes 19,555 seconds and MVMAE 19,601 seconds. The difference in training time is negligible. These results confirm that our "study-centric" approach adds significant clinical value and label efficiency without incurring a meaningful computational penalty. We will include this timing analysis in the revised manuscript to help practitioners assess the cost-benefit ratio of our method. Furthermore, while our structure-based supervision (MVMAE) introduces no meaningful penalty, adding the vision-to-text (V2T) objective does introduce the standard overhead associated with text decoders. However, as established in our previous responses, the V2T component is a modular extension; if training time or architectural complexity are primary constraints, the core MVMAE framework can be used independently. This distinction is consistent with broader literature [2], which demonstrates that while captioning-style objectives are more computationally intensive than simple alignment, their utility is context-dependent. By keeping these components decoupled, our method allows practitioners to choose the optimal trade-off between semantic richness in low-label regimes, and computational budget.
> >
> >
> > [2] Tschannen, M., Kumar, M., Steiner, A., Zhai, X., Houlsby, N., & Beyer, L. (2023). Image captioners are scalable vision learners too. Advances in Neural Information Processing Systems, 36, 46830-46855.
> >
> >
> > > MSE vs contrastive objectives
> >
> > We agree that this is a fair point; while we utilize an MSE loss for cross-view alignment, other supervisory signals such as contrastive objectives are indeed viable alternatives. Our choice of MSE was motivated by its simplicity and its success in recent masked autoencoding frameworks where the goal is to align representations in a high-dimensional latent space without the computational burden of managing large batch sizes or negative samples required by contrastive methods. Preliminary experiments showed little difference when exploring contrastive (InfoNCE) and MSE alignment losses, and we will discuss this in the updated manuscript.
> >
> >
> > > Elaborating on *Unknown* views
> >
> > We acknowledge the reviewer's concern regarding the impact of "Unknown" views on alignment quality. In our preliminary experiments, we found that the benefit of increased dataset size outweighed the potential noise introduced by including views with undefined projections.From a clinical standpoint, "Unknown" views often represent non-standard angles or poor-quality acquisitions that occur frequently in real-world practice; we believe that leveraging these samples allows the model to become more robust to the variability found in actual clinical settings. To quantify the prevalence of these views, we have compiled the following distribution across our pretraining datasets:
> >
> >
> > | Dataset        | Frontal (F) | Lateral (L) | Unknown (U) |
> > |----------------|-------------|-------------|-------------|
> > | MIMIC-CXR      | 239,931     | 116,555     | 15,465      |
> > | CheXpert+      | 191,071     | 32,391      | —           |
> > | PadChest       | 62,901      | 27,526      | 68,095      |
> > | Chest-Xray8    | 86,524      | —           | —           |
> >
> > As shown in the table, "Unknown" views represent a relatively small fraction of the data in high-quality sets like MIMIC-CXR. While PadChest has a higher proportion of unmapped views, we believe this is not significant given that the remaining two datasets do not report any unknown view. Effectively, the model learns that even if the specific projection is unclassified, it still belongs to the same multi-view study of that patient's anatomy. As a remark, the view information is only used for the view embedding in encoders and decoders, which results in a lightweight interaction.  We will update the manuscript to include these statistics and a discussion on why incorporating this "noisy" but high-volume data is beneficial for generalization.

---

> > > ### Author Response · Authors · 2026-01-23
> > > **Answer to reviewer DcLe [3/3]**
> > >
> > > > Visual interpretability
> > >
> > >
> > > We agree that visual interpretability is a powerful way to validate whether the model is truly capturing anatomically consistent regions across views. However, we have chosen not to include Grad-CAM maps in this version for two primary reasons. First, it is widely recognized in the medical AI community that interpretability tools like Grad-CAM are not yet "solved" and can often be misleading or produce "saliency maps" that do not strictly correlate with clinical decision-making. Second, our primary focus in this work was to establish a robust quantitative foundation for multi-view pretraining and its impact on label efficiency. We believe that a rigorous study of cross-view interpretability requires its own dedicated research path. We see this as a very interesting direction for future work; we have added a note to our discussion section to highlight this.
> > >
> > >
> > > > Time analysis
> > >
> > >
> > > We refer to the previous answer to this reviewer. As previously noted, the computational overhead of MVMAE is nearly identical to the standard approaches. To demonstrate this, we compared the wall-clock time for 500 epochs on 16 nodes: where the independent baseline takes 19,555s while MVMAE 19,601s. This shows that our structural supervision adds significant value with negligible computational penalty at pretraining, and since the decoders are discarded after pretraining, inference remains identical to a standard ViT. We have updated the manuscript with these timing and complexity discussion.
> > >
> > > > Contributions clarification
> > >
> > >
> > > Thanks for the pointer, we will update the introduction and conclusion of the manuscript to clarify the  framing of the contribution as a domain-specific synthesis of existing primitives.
> > >
> > >
> > > > General summary
> > >
> > >
> > > We hope that these responses and the updated manuscript address all your questions and concerns! Thank you for your constructive feedback, which has improved the clarity of our work.

---

### Author Response · Authors · 2026-01-23
**General answer to all reviewers**

We thank all reviewers for their thorough and constructive feedback. We appreciate the recognition of the clarity of the manuscript [**czM9**], the strong clinical motivation and our focus on exploiting the inherent structure of radiology studies [**ZDv9**,**czM9**,**DcLe**],  the simplicity and cleanliness of the proposed objectives [**ZDv9**], as well as the breadth of the experimental evaluation [**ZDv9**, **czM9**, **DcLe**]. We have included a detailed point-by-point response to all the reviewers' open questions and we are updating the manuscript accordingly. We will update it in Openreview as soon as all the experiments are finalized to include all updated tables.

One of the main open points concerns the vision-to-text (V2T) component. We clarify that V2T is a captioning-style objective intended to reflect the clinical reality in which CXR reports are narrative descriptions derived from visual evidence. While MVMAE-V2T does not consistently outperform the vision-only MVMAE in full-data settings, we found it noteworthy that in label-scarce regimes, where self-supervised pretraining plays a critical role, the V2T variant consistently achieves better downstream performance after full fine-tuning. We are revising the manuscript to explicitly highlight the performance differences between MVMAE-V2T and the vision-only model.

We are clarifying the main message to emphasize that our focus is on proposing a new way of structure-based supervision and not a novel self-supervised learning method. With regards to the experimental setting, we have included more extensive details of the training schedules and hyperparameters used, as well as time complexity insights of our model.  Finally, we will report the mean and standard deviation across three random seeds for all main results, and expand the discussion of limitations and scope. We thank the reviewers again for their valuable feedback, which has helped improve the clarity and rigor of the paper, and look forward to the next stage of the review process.

---

### Author Response · Authors · 2026-02-01
**General update to all reviewers**

Dear Reviewers,

thank you for your constructive comments, which helped us significantly improve the manuscript. We have carefully addressed your points and updated the paper accordingly. Below we summarize the main changes:

• **Multiple Seeds and Robustness of Results**

We re-ran the main experiments using three random seeds and now report the mean and standard deviation across seeds.
The results confirm the strong performance of MVMAE. When averaged across three random seeds, MVMAE reports the highest mean AUROC and F1 score across all labels on each dataset (see Table 1). Regarding the Top-5 results (see Table 2), MVMAE reports the highest AUROC on the combined dataset, while MVMAE-V2T variant reports the highest F1 score.

• **Correction on the Calibration Experiment**

We identified an error in the first version of the manuscript where calibration results were reported using full fine-tuning instead of linear probing, as intended. We have corrected this and now report proper linear probing results. MVMAE remains the best-calibrated model among all pretraining baselines. When compared to the supervised baseline, calibration performance is lower, which we believe is expected, as the supervised model is directly optimized for the classification task, whereas MVMAE is pretrained in a self-supervised manner and only adapted to the classification task in a second stage.

• **Clarified Contributions**

We clarified that our framework represents an adaptation of existing, well-established paradigms, specifically tailored to leverage the structural properties of medical data.

• **Generalized V2T Formulation**

We generalized the V2T formulation by introducing a weighting term $\gamma$ for the cross-entropy loss and explicitly state that $\gamma = 1$ is used in our experiments.

• **Related Work Update**

We added SigLIP-2 as a relevant work and discussed its key differences compared to MVMAE-V2T.

• **Dataset View Statistics**

We added detailed statistics on the view distribution for each dataset in the Appendix.

• **Training and Implementation Details**

We added an appendix section covering A timing analysis comparing the computational overhead introduced by cross-view alignment, additional discussion of the alignment objective, additional details on the learning rate and beta schedulers used.

• **Expanded Discussion and Conclusion**

We expanded the Discussion section with a clearer comparison between MVMAE and MVMAE-V2T, highlighting when and why each approach is beneficial, as well as their limitations. The Conclusion was updated accordingly to reflect these insights.

We thank you again for your valuable feedback, which helped improve the clarity, correctness, and completeness of the manuscript.

---

### Decision · Action_Editor_terM · 2026-03-07

**Recommendation:** Accept as is

**Audience:**

Yes

**Audience Explanation:**

The paper advances the state of the art for an important application area.

**Claims And Evidence:**

Yes

**Claims Explanation:**

The paper proposes a specialized self-supervised training pipeline for chest radiography by leveraging the structure of radiological studies. Its main contribution is MVMAE, along with a vision-to-text extension, MVMAE-V2T.

Reviewers generally appreciated the breadth of the evaluation across multiple datasets and baselines, as well as the clinical relevance.

The main criticism that remained unresolved after the rebuttal concerned novelty and framing. For instance, Reviewer DcLe commented that the work combines known ingredients such as MAE and multiview alignment, which the AE agrees with. However, this did not affect the positive assessment of correctness and rigor. One important concern was the need for greater rigor through evaluation across multiple seeds, which was addressed satisfactorily.

Not all comments were fully resolved during the rebuttal phase. For example, the evidence supporting the benefit of MVMAE-V2T remains mixed.

Reviewers who voted for rejection still evaluated claims as accurate and interesting for the community.

Overall, the paper makes an incremental but valuable and carefully evaluated contribution that will be useful to the community.